# Cold Rao-Blackwellized Straight-Through Gumbel-Softmax Gradient Estimator

## Abstract

The problem of estimating the gradient of an expectation in discrete random variables arises in many applications: learning with discrete latent representations, training neural networks with quantized weights, activations, conditional blocks, etc. This work is motivated by the development of the Gumbel-Softmax family of estimators, which are based on approximating argmax with a temperature-parametrized softmax. The state-of-the art in this family, the Gumbel-Rao estimator uses internal MC samples to reduce the variance and appars to improve with lower temperatures. We show that it possesses a zero temperature limit with a surprisingly simple closed form defining our new estimator called ZGR. It has favorable bias and variance properties, is easy to implement, computationally inexpensive and is obviously free of the temperature hyperparameter. Furthermore, it decomposed as the average of the straight through estimator and the DARN estimator – two basic but not very well performing on their own estimators. Unlike them, ZGR is shown to be unbiased for the class of quadratic functions of categorical variables. Experiments thoroughly validate the method.

## 1 Introduction

Discrete variables and discrete structures are important in machine learning. For example, the semantic hashing idea (Salakhutdinov & Hinton, 2009) is to learn a binary vector representation such that semantically similar instances (*e.g.* images or text documents) would have similar bit representations. This allows for a quick retrieval via a lookup table or via nearest neighbor search with respect to the Hamming distance of binary encodings. Recent work employs variational autoencoders (VAEs) with binary latent states (Shen et al., 2018; Dadaneh et al., 2020; Ñanculef et al., 2020). Another example is neural networks with discrete (binary or quantized) weights and activations. They allow for a low-latency and energy efficient inference, particularly important for edge devices. Recent results indicate that quantized networks can achieve competitive accuracy with a better efficiency in various applications (Nie et al., 2022).

VAEs and quantized networks are two diverse examples that motivate our development and the experimental benchmarks. Other potential applications include conditional computation (Bengio et al., 2013a; Yang et al., 2019; Bulat et al., 2021) reinforcement learning (Yin et al., 2019), learning task-specific tree structures for agglomerative neural networks (Choi et al., 2018), neural architecture search (Chang et al., 2019) and more.

The learning problem in the presence of stochastic variables is usually formulated as minimization of the expected loss. The gradient-based optimization requires a gradient of the expectation in the probabilities of random variables (or parameters of networks inferring those probabilities). Unbiased gradient estimators have been developed (Williams, 1992; Grathwohl et al., 2018; Tucker et al., 2017; Gu et al., 2016). These estimators work even for non-differentiable losses, however their high variance is the main limitation. More recent advances (Yin et al., 2019; Kool et al., 2020; Dong et al., 2020; 2021; Dimitriev & Zhou, 2021b;a) reduce the variance by using several cleverly coupled samples. However, the hierarchical (or deep) case is not addressed satisfactory. There is experimental evidence that the variance grows substantially with the depth of the network leading to very poor performance in training (*e.g.*, Shekhovtsov et al. 2020, Fig.C.6, C.7). Furthermore, existing extensions of coupled sampling methods to networks with $L$ dependent layers (Dong et al., 2020; Yin et al., 2019) apply their base method in every layer, requiring several complete forward

**Table 1:** Computation complexity of estimators in a hierarchical network with $L$ dependency layers of $K$-way categorical variables. All methods require a backward pass. $^*$ – educated guess.

| Method | Unbiased | Forward passes | Cost per cat. variable per pass |
|---|---|---|---|
| ST, GS, GS-ST, ZGR | ✗ | 1 | $O(K)$ |
| GR-MC($M$) | ✗ | 1 | $O(MK)$ |
| REINFORCE | ✓ | 1 | $O(K)$ |
| RF($M$) (Kool et al., 2019) | ✓ | $M \geq 2$ | $O(K)$ |
| ARSM (Yin et al., 2019) | ✓ | $O(K^2L^2)$ | $O(1)$ |
| DisARM-* (Dong et al., 2020; 2021) | ✓ | $O(2L^2)$ | $O(K)$ |
| CARMS($M$) Dimitriev & Zhou (2021b;a) | ✓ | $O(ML^2)^*$ | $O(M^2K^2)^*$ |

samples per layer. The computation complexity thus grows quadratically with the number of layers, as summarized in Table 1.

A different family of methods, fitting practical needs for deep models better, exploits continuation arguments. It includes ST variants (Bengio et al., 2013b; Shekhovtsov & Yanush, 2021; Pervez et al., 2020) and Gumbel-Softmax variants (to be discussed below). These methods assume the loss function to be differentiable and try to estimate the derivative with respect to parameters of a discrete distribution from the derivative of the loss function. Such estimators can be easily incorporated into back-propagation by adjusting the forward and backward passes locally for every discrete variable. They are, in general, biased because the derivative of the loss need not be relevant for the discrete expectation. The rationale though is that it may be possible to obtain a low variance estimate at a price of small bias, *e.g.* for a sufficiently smooth loss function (Shekhovtsov & Yanush, 2021).

Gumbel Softmax (Jang et al., 2017) and the concurrently developed Concrete relaxation (Maddison et al., 2017) enable differentiability through discrete variables by relaxing them to real-valued variables with a distribution approximating the original discrete distribution. The tightness of the relaxation is controlled by the temperature parameter $t > 0$. The bias can be reduced by decreasing the temperature, but the variance grows as $O(1/t)$ (Shekhovtsov, 2021). Gumbel-Softmax Straight-Through (GS-ST) heuristic (Jang et al., 2017) uses discrete samples on the forward pass, but relaxed ones on the backward, reducing side biases (see below). The Gumbel-Rao (GR) estimator (Paulus et al., 2021) is a recent improvement of GS-ST, which can substantially reduce its variance by a local expectation. However the local expectation results in an intractable integration in multiple variables, which is approximated by sampling. The experiments (Paulus et al., 2021) suggest that this estimator performs better at lower temperatures, which requires more MC samples. Therefore the computation cost appears to be a major limitation.

Inspired by the performance of GR at low temperatures, we analyze its behavior for temperatures close to and in the limit of the absolute zero. Note that because the underlying relaxation becomes discrete in the cold limit, leading to explosion of variance in both GS and GS-ST, they do not have a zero temperature limit. It is not obvious therefore that GR would have one. We prove that it does and denote this limit estimator as ZGR. In the case of binary variables we give an asymptotic series expansion of GR around $t = 0$ and show that ZGR has a simple analytic expression, matching the already known DARN($\frac{1}{2}$) estimator by Gregor et al. (2014) (also re-discovered as importance reweighed ST by Pervez et al. 2020). In the general categorical case, we obtain the analytic expression of ZGR and show that it is a new estimator with a simple formula $\frac{1}{2}$(ST+DARN). We show that ZGR is unbiased for all quadratic functions of categorical variables and experimentally show that it achieves a useful bias-variance tradeoff. We also contribute a refined experimental comparison of GS family with unbiased estimators.

## 2 BACKGROUND

Let $x$ be a categorical random variable taking values in $\mathcal{K} = \{0, \dots K - 1\}$ with probabilities $p(x; \eta)$ parametrized by $\eta$. Let $\phi(x) \in \mathbb{R}^d$ be an *embedding* of the discrete state $x$ in the vector space. Categorical variables are usually represented using 1-hot embedding, in this case $d = K$. The value of $x$ itself will still be used as an index. For binary and quantized variables we will adopt the embedding $\phi(x) = x$, in this case $d = 1$.

Let $\mathcal{L}\colon \mathbb{R}^d \to \mathbb{R}$ be a differentiable loss function. We will denote the Jacobian of $\mathcal{L}$ with respect to an input $\phi$ as $J_\phi = \frac{\mathrm{d}\mathcal{L}}{\mathrm{d}\phi}$. It is the transposed gradient of $\mathcal{L}$ with respect to $\phi$, nevertheless, we will informally refer to such Jacobians as gradients. For brevity, let us also use a shorthand $\mathcal{L}(x) = \mathcal{L}(\phi(x))$. The goal is to estimate the gradient of the expected loss

$$J_\eta = \tfrac{\mathrm{d}}{\mathrm{d}\eta} \mathbb{E}[\mathcal{L}(x)] = \tfrac{\mathrm{d}}{\mathrm{d}\eta} \sum_x \mathcal{L}(x) p(x;\eta). \tag{1}$$

In a network with many (dependent) categorical variables it has proven efficient (*e.g.*, Bengio et al. 2013a; Jang et al. 2017; Paulus et al. 2021; Shekhovtsov & Yanush 2021; Pervez et al. 2020) to consider estimators that make an estimate of $J_\eta$ based on the gradient $J_\phi = \frac{\mathrm{d}\mathcal{L}(\phi(x))}{\mathrm{d}\phi}$ at a sample $x$ from $p(x;\eta)$ in (1). Such estimators can be easily extended to losses in multiple discrete variables (*e.g.* defined by a stochastic computation graph) by simply applying the elementary estimator whenever the respective Jacobian is needed in backpropagation. In this case $J_\eta$ is the gradient in a specific variable or intermediate activation $\eta$ at a joint sample.

Let us review the basic estimators with which we will work theoretically.

**REINFORCE** We can use the log-derivative trick to rewrite

$$\tfrac{\mathrm{d}}{\mathrm{d}\eta} \sum_x \mathcal{L}(x) p(x;\eta) = \sum_x \mathcal{L}(x) \tfrac{\mathrm{d}p(x;\eta)}{\mathrm{d}\eta} = \sum_x \mathcal{L}(x) p(x;\eta) \tfrac{\mathrm{d}\log p(x;\eta)}{\mathrm{d}\eta} = \mathbb{E}\big[\mathcal{L}(x) \tfrac{\mathrm{d}\log p(x;\eta)}{\mathrm{d}\eta}\big] \tag{2}$$

and define the REINFORCE estimate (Williams, 1992) to be $J_\eta^{\mathrm{RF}} = \mathcal{L}(x) \tfrac{\mathrm{d}\log p(x;\eta)}{\mathrm{d}\eta}$, where $x \sim p(x;\eta)$. This estimator is clearly unbiased as $\mathbb{E}[J_\eta^{\mathrm{RF}}] = J_\eta$, but may have a high variance.

**ST** Let $\bar{\phi}(\eta) = \mathbb{E}[\phi(x)] = \sum_x \phi(x) p(x;\eta)$ – the *mean embedding* in $\mathbb{R}^d$ under the current distribution of $x$. The Straight-Through estimator is

$$J_\eta^{\mathrm{ST}} = J_\phi \tfrac{\mathrm{d}\bar{\phi}(\eta)}{\mathrm{d}\eta}. \tag{3}$$

Note, ST estimators of different empirical forms exist. The present definition for categorical variables is the same as, *e.g.*, by Gu et al. (2016) and is consistent with Hinton (2012) and Shekhovtsov & Yanush (2021) in the binary case.

**DARN** Gregor et al. (2014) use LR with a baseline $b(x) = \mathcal{L}(x) + J_\phi(\bar{\phi} - \phi(x))$, which is a first order Taylor approximation of $\mathcal{L}$ about $\phi(x)$ evaluated at some point $\bar{\phi}$. This results in the estimator

$$J_\eta^{\mathrm{DARN}(\bar{\phi})} = J_\phi(\phi(x) - \bar{\phi}) \tfrac{\mathrm{d}\log p(x;\eta)}{\mathrm{d}\eta}. \tag{4}$$

When $x$ is binary, the choice $\bar{\phi} = \frac{1}{2}\sum_x \phi(x)$ – the mean embedding under the uniform distribution, ensures that the estimator is unbiased for any quadratic function. However, for categorical variables no $\bar{\phi}$ with such property exist. Gu et al. (2016) have experimentally tested several heuristic choices for $\bar{\phi}$, including $\bar{\phi} = \bar{\phi}(\eta)$, and found that none performed well in the categorical case.

**GS** The Gumbel-Softmax (GS) estimator (Jang et al., 2017) is a relaxation of the Gumbel-argmax sampling scheme. Let $\theta_k = \log p(x{=}k;\eta)$. Let $G_k \sim \mathrm{Gumbel}(0,1)$, $k \in \mathcal{K}$, where $\mathrm{Gumbel}(0,1)$ is Gumbel distribution with cdf $F(u) = e^{-e^{-u}}$. Then

$$x = \arg\max_k(\theta_k + G_k) \tag{5}$$

is a sample from $p(x;\eta)$. The relaxation is obtained by using a $\mathrm{softmax}$ instead of $\arg\max$. This construction assumes one-hot embedding $\phi$ and creates relaxed (continuous) samples in the simplex $\Delta^K$. Formally, introducing temperature hyperparameter $t$, it can be written as

$$\tilde{\phi} = \mathrm{softmax}((\theta + G)/t) =: \mathrm{softmax}_t(\theta + G); \tag{6a}$$

$$J_\theta^{\mathrm{GS}} = \tfrac{\mathrm{d}\mathcal{L}(\tilde{\phi})}{\mathrm{d}\theta} = \tfrac{\mathrm{d}\mathcal{L}(\tilde{\phi})}{\mathrm{d}\tilde{\phi}} \tfrac{\mathrm{d}\tilde{\phi}}{\mathrm{d}\eta}. \tag{6b}$$

There are two practical concerns. First, the loss function is evaluated at a relaxed sample, which in a large computation graph can offset estimates of all other gradients even not related to the discrete variable we relax. This effect can be mitigated by using a smaller temperature, causing relaxed samples $\tilde{\phi}$ to concentrate in a corner of the simplex. However, and this is the second concern, the variance of the estimator grows as $O(\frac{1}{t})$ if $t$ is decreased towards zero (Shekhovtsov, 2021).

**GS-ST** The Straight-Through Gumbel-Softmax estimator (Jang et al., 2017) is an empirical modification of GS, addressing the first concern above. It uses discrete samples in the forward pass but swaps in the Jacobian of the continuous relaxation in the backward pass:

$$G_k \sim \text{Gumbel}(0,1), \ k \in \mathcal{K}; \tag{7a}$$

$$x = \arg\max_k(\theta_k + G_k); \tag{7b}$$

$$\tilde{\phi} = \text{softmax}_t(\theta + G); \tag{7c}$$

$$J_\theta^{\text{GS-ST}} = \frac{\mathrm{d}\mathcal{L}(\phi(x))}{\mathrm{d}\phi}\frac{\mathrm{d}\tilde{\phi}}{\mathrm{d}\theta}. \tag{7d}$$

Notice that the hard sample $\phi(x)$ and the relaxed sample $\tilde{\phi}$ are entangled through $G$. Although, $x$ has the law of $p(x;\eta)$ as desired, not biasing other variables, there is still bias in estimating the gradient in $\theta$, which is typically larger than that of GS (see Fig. 1). To make the bias smaller the temperature $t$ should be decreased, however, the variance still grows as $O(1/t)$ (Shekhovtsov, 2021). Values of $t$ between $0.1$ and $1$ are used in practice (Jang et al., 2017).

**GR** Notice that the forward pass in GS-ST is fully determined by $x$ alone and the value of $G$ that generated that $x$ is needed only in the backward pass. Paulus et al. (2021) proposed that the variance of GS-St can be reduced by computing the conditional expectation in $G|x$, leading to the Gumbel-Rao estimator:

$$J_\theta^{\text{GR}} = \mathbb{E}_{G|x}\left[J_\theta^{\text{ST-GS}}(G)\right] = \frac{\mathrm{d}\mathcal{L}(\phi(x))}{\mathrm{d}\phi}\mathbb{E}_{G|x}\left[\frac{\mathrm{d}\tilde{\phi}}{\mathrm{d}\theta}\right]. \tag{8}$$

Because the value of the loss $\mathcal{L}(x)$ and its gradient do not depend on the specific realization of $G|x$, enabling the equality above, the expectation is localized and can be computed in the backward pass. However, this expectation is in multiple variables and is not analytically tractable. Paulus et al. (2021) use Monte Carlo integration with $M$ samples from $G|x$. In their experiments they report improvement of the mean squared error of the estimator when the temperature was decreasing from $1$ down to $0.1$. The trend suggests that it would improve even further below $t = 0.1$ provided that the conditional expectation is approximated accurately enough, indicating that the variance does not grow with the decrease of temperature in contrast to $O(1/t)$ asymptote for GS and GS-ST and so there might be a meaningful cold limit.

## 3 METHOD

Given the experimental evidence about the GR estimator, we took the challenge to study its cold asymptotic behavior, *i.e.* for $t \to 0$. The temperatured softmax in (7c) approaches a non-differentiable $\arg\max$ indicator in this limit and we have to handle the limit of the GR estimator with care to obtain correct results. We first analyze the binary case, where derivations are substantially simpler. Proofs of all formal claims can be found in Appendix A.

### 3.1 BINARY CASE

In the case with two categories we can simplify the initial ST-GS estimator as follows. We assume $x \in \{0,1\}$ and $\phi(x) = x$. The argmax trick can be expressed as $x = [\![\theta_1 + G_1 \geq \theta_0 + G_0]\!]$, where $[\![\cdot]\!]$ is the Iverson bracket. It is convenient to assume that the distribution of $x$ is parametrized so that $p(x{=}1;\eta) = \sigma(\eta)$, the logistic sigmoid function. This is without loss of generality, because any other parametrization will result in just an extra deterministic Jacobian. Recalling that $\theta_k = \log p(x{=}k;\eta)$, we have $\theta_1 - \theta_0 = \eta$. Next, denoting $Z = G_1 - G_0$, we can write the argmax trick compactly as $x = [\![\eta + Z \geq 0]\!]$. Being the difference of two Gumbel(0,1) variables $Z$ follows the standard logistic distribution (with cdf $\sigma(z)$). The GR estimator of gradient in $\eta$ simplifies as

$$Z \sim \text{Logistic}(0,1); \quad x = [\![\eta + Z \geq 0]\!]; \quad \tilde{x} = \sigma_t(\eta + Z); \quad J_\eta^{\text{GR}} = \frac{\mathrm{d}\mathcal{L}(x)}{\mathrm{d}x}\mathbb{E}_{Z|x}\left[\frac{\mathrm{d}\tilde{x}}{\mathrm{d}\eta}\right], \tag{9}$$

where $\sigma_t(u) = \sigma(u/t)$ is the temperatured logistic sigmoid function. Although there is no closed form, we can compute, with a careful limit-integral exchange, the series expansion around $t = 0$.

**Proposition 1.**

$$J_\eta^{\text{GR}} = \frac{\mathcal{L}'(x)}{p(x)}p_Z(\eta)\left(\frac{1}{2} + (2x-1)c_1\log(2)t\right) + O(t^2), \tag{10}$$

where $p_Z$ is the logistic density: $p_Z(\eta) = \sigma(\eta)\sigma(-\eta)$ and $c_1 = 2p - 1$.

**Corollary 1.** In the limit $t \to 0$ the GR estimator becomes the DARN($\frac{1}{2}$) estimator.

This establishes an interesting theoretical link, but discovers no new method for practical applications, as the DARN estimator is already known. Using the same expansion, we can study the asymptotic bias and variance of GR around $t = 0$.

**Corollary 2.** The mean and variance of the GR estimator (9) in the asymptote $t \to 0$ are:

$$\mathbb{E}[J_\eta^{\text{GR}}] = p(1-p)\Big(\tfrac{1}{2}(\mathcal{L}'(1) + \mathcal{L}'(0)) + (\mathcal{L}'(1) - \mathcal{L}'(0))\tilde{c}_1 t\Big) + O(t^2), \qquad (11a)$$

$$\mathbb{V}[J_\eta^{\text{GR}}] = (p(1-p))^3\Big(\tfrac{1}{4}(a-b)^2 + \tfrac{1}{2}(a^2 - b^2)\tilde{c}_1 t\Big) + O(t^2), \qquad (11b)$$

where $p = \sigma(\eta)$, $a = \frac{\mathcal{L}'(1)}{p}$, $b = \frac{\mathcal{L}'(0)}{1-p}$ and $\tilde{c}_1 = (2p-1)\log(2)$.

This allows to make some predictions, in particular in the case of a linear objective $\mathcal{L}$. In this case the bias is $O(t^2)$ and the squared bias is $O(t^4)$. Therefore the MSE is determined by the variance alone up to $O(t^4)$. The dependence of variance on $t$ for a linear objective is negative in the first order term. Therefore the temperature corresponding to the minimum MSE will be non-zero.

## 3.2 General Categorical Case

In the general categorical case, the analysis is more complicated (exchange of the limit and multivariate integral over $G|x$) but gives a novel result:

**Theorem 1** (ZGR). *The Gumbel-Rao estimator for one-hot embedding $\phi$ in the limit of zero temperature is given by*

$$J_{\theta_i}^{ZGR} = \begin{cases} \tfrac{1}{2}(J_{\phi_i} - J_{\phi_x})p(x{=}i; \eta) & \text{if } i \neq x; \\ -\tfrac{1}{2}\sum_{j \neq x}(J_{\phi_j} - J_{\phi_x})p(x{=}j; \eta) & \text{if } i = x. \end{cases} \qquad (12)$$

**Proposition 2.** ZGR estimator decomposes as

$$\boxed{J^{\text{ZGR}} = \tfrac{1}{2}(J^{\text{ST}} + J^{\text{DARN}(\bar{\phi}(\eta))}),} \qquad (13)$$

*i.e.*, with the choice $\bar{\phi} = \bar{\phi}(\eta)$ in DARN.

This is consistent with the expression for the binary case given by Corollary 1 by noting that in this case $(\text{ST} + \text{DARN}(\bar{\phi}(\eta)))/2$ is exactly DARN(1/2). As we have defined ST and DARN estimators for a general embedding, the form (2) is a valid expression of ZGR for any embedding (a change of the embedding is just a linear transform). This form shows a surprising connection. Neither ST nor DARN estimators perform particularly well in categorical VAEs on their own (Gu et al., 2016; Paulus et al., 2021). However ZGR, being effectively their average, appears superior. It has the following property, truly extending the design principle of binary DARN($\frac{1}{2}$) to the categorical case:

**Theorem 2.** *ZGR is unbiased for quadratic loss functions $\mathcal{L}$ with arbitrary coefficients.*

Both ST and DARN($\bar{\phi}(\eta)$) are unbiased for linear functions and the theorem shows their biases for quadratic functions are exactly opposite and cancel in the average. Because the bias of ST-GS, GR and GR-MC is the same and we have shown that ZGR is the limit of GR at $t \to 0$, the following is straightforward.

**Corollary 3.** ST-GS and GR are asymptotically unbiased for quadratic functions with $t \to 0$.

These results extend to multiple independent discrete variables as follows.

**Corollary 4.** Let $x_1, \ldots, x_n$ be independent categorical variables and $\mathcal{L}(x_1, \ldots, x_n)$ be such that for all $i$ and all configurations $x$ the restriction $\mathcal{L}(x_i)$ is a quadratic function. Then ZGR is unbiased.

The unbiased property for quadratic function gives us some intuition about applicability limits of ZGR. Namely, if the loss function is reasonably smooth, such that it can be approximated well by a quadratic function, we expect gradient estimates to be accurate. Compared to ST, which is unbiased for multilinear functions only, we hypothesize that ZGR can capture interactions more accurately.

## 4 EXPERIMENTS

We will compare ZGR with Gumbel-Softmax (**GS**), Straight-through Gumbel-Softmax (**GS-ST**) (Jang et al., 2017), Gumbel-Rao with MC samples (**GR-MC**) (Paulus et al., 2021) and the **ST** estimator (3). We also compare to the REINFORCE with the leave-one-out baseline (Kool et al., 2019) using $M \geq 2$ inner samples, denoted RF($M$), which is a strong baseline amongst unbiased estimators. In some tests we include **ARSM** (Yin et al., 2019), which requires more computation than RF(4) but performs worse. See Appendix B.2 for details of implementations.

### 4.1 DISCRETE VARIATIONAL AUTOENCODERS

We follow established benchmarks for evaluating gradient estimators in discrete VAEs. We use a MNIST data with a fixed binarization (Yin et al., 2019) and Omniglot data with dynamic binarization (Burda et al., 2016; Dong et al., 2021). We use the encoder-decoder networks same as (Yin et al., 2019; Dong et al., 2021) up to the following difference. We embed categorical variables with $2^b$ states as $\{-1, 1\}^b$ vectors. This allows to vary the number of categorical variables and categories while keeping the decoder size the same. Please see full details in Appendix B.

#### 4.1.1 ZERO TEMPERATURE LIMIT AND ESTIMATION ACCURACY IN VAE

First, we measure the gradient estimation accuracy at a particular point of VAE training, comparing GS family of estimators at different temperatures as in (Paulus et al., 2021, Fig 2b.). This referenced plot shows a steady decrease of MSE with the decrease of temperature down to $0.1$ and we were expecting ZGR to achieve the lowest MSE.

As an evaluation point we take a VAE model after 100 epochs of training with RF(4). We then measure the bias and variance of all gradient estimators as follows. The loss function is the average ELBO of a fixed random mini-batch of size 200. Let $X \in \mathbb{R}^d$ be the reference unbiased estimator and $Y \in \mathbb{R}^d$ be the tested estimator. We want to measure the average over parameters ($=$ dimensions of the gradient) squared bias, which can be written as $b^2 = \frac{1}{d}\|\mathbb{E}[X] - \mathbb{E}[Y]\|^2$. We obtain $n_1 = 10^4$ independent samples $X_i$ from RF(4) and $n_2 = 10^4$ independent samples $Y_i$ from the tested estimator and compute an unbiased estimate of $b^2$:

$$\widehat{b^2} = \frac{1}{d}\|\hat{\mu}_1 - \hat{\mu}_2\|^2 - \frac{V_1}{n_1} - \frac{V_2}{n_2}, \tag{14}$$

where $\hat{\mu}_1$ is the sample mean of $X$ and $V_1$ is the average (over dimensions) sample variance of $X$ and $\hat{\mu}_2$ and $V_2$ are likewise for $Y$. The variance of the evaluated estimator is just $V_2$.

The results are shown in Fig. 1. All of ST-GS and GR estimators share the same bias according to the theory but differ in variance. GS estimator is asymptotically unbiased but the variance grows as $O(1/t)$. We observe that the variance is by several orders larger than the squared bias. Respectively, the mean squared error, which is the sum of the variance and the squared bias, is dominated by the variance alone for all methods. This is in high contradiction with MSE analysis in (Paulus et al., 2021, Fig 2b.), which we deem incorrect.

Note however, that the effect of the bias-variance trade-off in learning is not straightforward: common optimization methods use momentum as an effective way of variance reduction while the bias can be correlated across iterations and may potentially accumulate. Therefore MSE is not necessarily indicative of performance and ZGR still fulfills our expectations of the zero limit estimator: it has the limiting bias, which is the lowest in the GR/GS-ST family, and the limiting variance which is moderate. In particular the variance is better than that of GR for temperatures below about $0.1$ and is comparable to the variance of RF(4). More tests at different stages of training are provided in Appendix B.4.

#### 4.1.2 TRAINING PERFORMANCE IN VAE

Next we compare training performance of several methods, in the same setting as prior work. In particular we use the same Adam optimizer, batch size, learning rate and training duration as (Dong et al., 2021; Dimitriev & Zhou, 2021b). Full details are given in Appendix B.3.

Table 2 presents results for two datasets and different splitting of latent bits into discrete variables (from binary to 64-way categorical). We observe the following: 1) ST performs the worst and we

**Table 2:** VAE training negative ELBO for binary MNIST $> 0.5$ and the down-sampled and dynamically binarized Omniglot. Each value is the mean over 3 random initializations and confidence intervals are $\pm(\max - \min)/2$ of the 3 runs. Bold results are the three best ones per configuration.

| | Method | \multicolumn{4}{c}{Number of **C**ategories & Categorical **V**ariables} |
| | | C2 V192 | C4 V96 | C16 V48 | C64 V32 |
|---|---|---|---|---|---|
| **MNIST-B** | GS(t=0.1) | 91.2±0.1 | 84.4±0.5 | 82.8±0.5 | 86.9±0.9 |
| | GS-ST(t=0.1) | 92.0±0.4 | 85.4±0.2 | 84.1±0.5 | 90.0±0.5 |
| | GR-MC(t=0.1,M=10) | 88.8±0.2 | **82.7±0.1** | **81.0±0.1** | **82.4±0.1** |
| | GR-MC(t=0.1,M=100) | **88.0±0.5** | 82.4±0.2 | 80.5±0.3 | **81.7±0.5** |
| | ZGR | **88.6±0.2** | 83.0±0.1 | **80.6±0.4** | **81.9±0.1** |
| | ST | 105.0±0.2 | 105.4±0.1 | 106.0±0.1 | 106.7±0.1 |
| | RF(M=2) | 92.1±0.4 | 86.3±0.5 | 88.6±0.5 | 96.7±0.1 |
| | RF(M=4) | **88.2±0.3** | **82.7±0.1** | 82.2±0.5 | 87.2±0.7 |
| **Omniglot-28-D** | GS(t=0.1) | 117.6±0.2 | 116.0±0.2 | 117.0±0.1 | 120.6±0.2 |
| | GS-ST(t=0.1) | 119.4±0.1 | 116.7±0.2 | 118.5±0.1 | 123.2±0.3 |
| | GR-MC(t=0.1,M=10) | **116.6±0.2** | **114.6±0.1** | **115.7±0.1** | **117.8±0.2** |
| | GR-MC(t=0.1,M=100) | **116.5±0.1** | **114.4±0.1** | **115.2±0.1** | **116.9±0.3** |
| | ZGR | **116.6±0.4** | **114.5±0.1** | **115.4±0.1** | **117.0±0.2** |
| | ST | 130.2±0.1 | 130.5±0.1 | 131.5±0.2 | 132.0±0.1 |
| | RF(M=2) | 120.5±0.4 | 118.8±0.1 | 122.0±0.3 | 127.5±0.4 |
| | RF(M=4) | 117.1±0.1 | 115.8±0.2 | 117.9±0.1 | 120.7±0.1 |

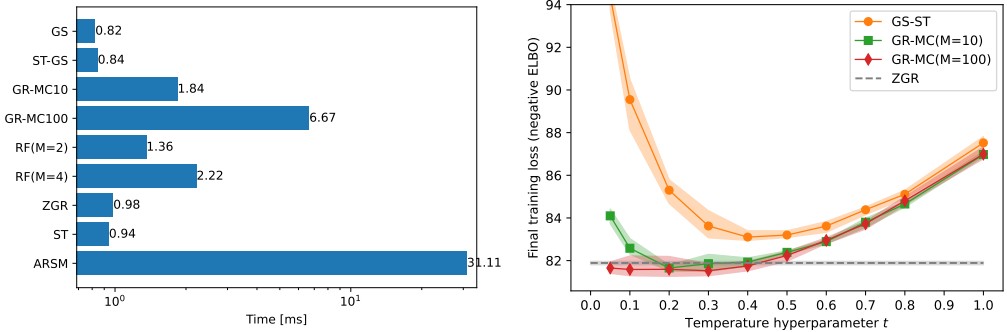

**Figure 1:** Gradient estimation accuracy in VAE on MNIST-B. Average (per parameter) squared bias (*left*) and variance (*right*) of gradient estimators versus temperature at a model snapshot after 100 epochs of training with RF(4). Confidence intervals are $95\%$ empirical intervals of 100 bootstrap samples of the estimates (very narrow for variances).

**Figure 2:** *Left:* time [ms] of a forward-backward pass on GPU (16 variables, 16 categories, single batch of size 200, MNIST-B). GS and GS-ST estimators are shipped with Pytorch while other estimators are implemented by us using tensor operations in Pytorch (no loops). ZGR is significantly faster than GR-MC and one of the fastest methods overall. *Right:* Training ELBO after 500 epochs of training with different choices of temperature (MNIST-B, 32 variables 64 categories). The regret of ZGR w.r.t. GR-MC at the optimal temperature is insignificant. The plot shows the mean from 5 random initializations with confidence intervals $\pm(\max - \min)/2$ of the 5 runs.

blame its high bias (*c.f.* Fig. 1) 2) ZGR performs no worse than GR-MC variants. In Fig. 2 we additionally verify that at no other temperature GR-MC can achieve significantly better results; 3) ZGR outperforms RF(2) and RF(4), significantly so with more categories. Finally, we measure the (orientational) computation time in Fig. 2 and observe that ZGR is faster than both GR-MC and RF(2), consistently with theoretical expectations in Table 1.

Finally, according to the published results in a similar setup, the recent unbiased methods (Dong et al., 2021; Dimitriev & Zhou, 2021b) appear to improve only marginally over RF with an equal number of samples, *i.e.* the difference is much smaller than between ZGR and RF(2). We leave a peer-to-peer comparison to future work.

## 4.2 QUANTIZED NEURAL NETWORKS

The mainstream progress in training quantized and binary neural networks, following Hubara et al. (2017), has been achieved so far using empirical variants of ST estimator (with different clamping rules, *etc.*) applied to deterministically quantized models, where there is no gradient to be estimated in the first place. A sound training approach is to consider a stochastic relaxation, replacing all discrete weights and activations by discrete random variables, leading in the binary case to stochastic binary networks (Peters & Welling, 2018; Roth et al., 2019; Shekhovtsov & Yanush, 2021).

We will consider a parameter-efficient stochastic relaxation for quantization of Louizos et al. (2019). In this model the distribution of a quantized weight or activation $x$ is defined by a single real-valued input $\eta$ via: $x = \lfloor \eta + z \rceil_{\mathcal{K}}$, where $\lfloor \cdot \rceil$ rounds to the nearest integer in $\mathcal{K}$ and $z$ is an injected noise, such as logistic noise. Therefore $x$ is a discrete integer variable with a distribution determined by $\eta$. In Appendix B.5 we give an comprehensive evaluation of bias and variance of estimators for a single such quantization unit. In a deep network, the pre-activation input $\eta$ depends on the weights of the current layer as well as on the preceding activations (both stochastically quantized), causing a hierarchical dependence.

We train a convolutional network, closely following Louizos et al. (2019) and test on MNIST and FashionMNIST. We do not quantize the input (it has 8 bit resolution in the dataset), the first and last weight matrices are quantized to 4 bits. All inner layer weights and activations are quantized to 2 bits or below. We form two real-valued baselines applying ReLU or Clamp ($x \mapsto \min(\max(x, 0), K - 1)$) as activations instead of quantization. We expect that with enough bits, quantized training should achieve performance of at least the Clamp variant. We evaluate training with logistic injected noise (as in Louizos et al. (2019)) and *triangular* noise with the density $p(z) = \max(0, 1 - |z|)$. For GS-ST variants we enable high temperatures (0.5, 1, 2) as recommended by Louizos et al. (2019). See Appendix B.5 for details of the experimental setup.

The results are presented in Table 3. We see that the best results for are obtained with estimators having the least variance, prominently ST and GS-ST($t$=2). It suggests that the bias is less detrimental in this application. In Fig. 3 we measured bias and variance along the training trajectory of ZGR following the same methodology as in Section 4.1.1 with $10^4$ samples from the reference RF(4) estimator and $10^3$ samples from candidate estimators. The two methods with the lowest variance are exactly ST and GS-ST($t$=2) while the bias was hard to measure accurately to draw any conclusions. More generally, the ranking of results in Table 3 is quite similar to the ranking of variance in Fig. 3. In particular, variance of RF(4) is several orders larger than that of biased estimators and its test accuracy is completely out of the competition.

Regarding performance of ZGR we observe the following: 1) it outperforms GR-MC with temperature 0.1, more clearly so on Fashion-MNIST (Table B.2), while being cheaper and simpler; 2) It is close in performance to the best results obtained with more biased estimators.

## 5 DISCUSSION / CONCLUSION

On the theoretical side, we showed that GR estimator has a zero temperature limit, computed this limit, studied its properties and connected to the existing estimators. Despite we derived ZGR from the Gumbel-Softmax family, we do not consider it to be a proper member of this family. The straight-through heuristic in GS-ST disposes of relaxed samples on the forward pass. The zero temperature limit disposes of them also on the backward pass, leaving essentially nothing form the Gumbel-

**Table 3:** MNIST classification test error[%] in deterministic mode (no injected noises at test time) for different bit-width per weigh and activation (T denotes ternary). Hyperparameters are selected on the validation set. Reference test errors: ReLU $0.69\%$ , Clamp $0.64\%$.

| | | Weights [bits] / Activations [bits] | | | |
|---|---|---|---|---|---|
| | Method | 2/2 | T/T | T/1 | 1/1 |
| **Logistic Noise** | GS-ST(t=2) | $\mathbf{0.75} \pm 0.03$ | $\mathbf{0.73} \pm 0.05$ | $\mathbf{0.74} \pm 0.04$ | $\mathbf{0.83} \pm 0.05$ |
| | GS-ST(t=1) | $\mathbf{0.75} \pm 0.03$ | $0.78 \pm 0.08$ | $0.88 \pm 0.11$ | $\mathbf{0.86} \pm 0.04$ |
| | GS-ST(t=0.5) | $0.84 \pm 0.05$ | $0.86 \pm 0.04$ | $0.91 \pm 0.11$ | $1.02 \pm 0.08$ |
| | GR-MC(t=0.5,M=10) | $\mathbf{0.71} \pm 0.06$ | $\mathbf{0.78} \pm 0.06$ | $\mathbf{0.88} \pm 0.10$ | $0.95 \pm 0.09$ |
| | GR-MC(t=0.1,M=10) | $0.82 \pm 0.04$ | $0.85 \pm 0.07$ | $0.96 \pm 0.04$ | $1.14 \pm 0.04$ |
| | ZGR | $0.84 \pm 0.09$ | $0.86 \pm 0.07$ | $0.95 \pm 0.06$ | $1.07 \pm 0.06$ |
| | ST | $0.76 \pm 0.04$ | $\mathbf{0.77} \pm 0.05$ | $\mathbf{0.76} \pm 0.01$ | $\mathbf{0.81} \pm 0.02$ |
| | RF(M=2) L | 38.9 | 35.9 | 51.0 | 52.3 |
| | RF(M=4) L | 31.6 | 39.8 | 35.6 | 55.4 |
| **Triangular Noise** | GS-ST(t=2) | $0.73 \pm 0.05$ | $\mathbf{0.72} \pm 0.07$ | $0.76 \pm 0.05$ | $\mathbf{0.85} \pm 0.06$ |
| | GS-ST(t=1) | $\mathbf{0.71} \pm 0.06$ | $0.77 \pm 0.03$ | $\mathbf{0.74} \pm 0.03$ | $0.90 \pm 0.07$ |
| | GS-ST(t=0.5) | $0.73 \pm 0.04$ | $0.84 \pm 0.07$ | $0.81 \pm 0.04$ | $0.88 \pm 0.11$ |
| | GR-MC(t=0.5,M=10) | $\mathbf{0.70} \pm 0.06$ | $0.76 \pm 0.06$ | $\mathbf{0.71} \pm 0.05$ | $\mathbf{0.83} \pm 0.05$ |
| | GR-MC(t=0.1,M=10) | $0.78 \pm 0.01$ | $\mathbf{0.75} \pm 0.07$ | $0.80 \pm 0.03$ | $0.98 \pm 0.02$ |
| | ZGR | $0.75 \pm 0.05$ | $0.78 \pm 0.04$ | $0.80 \pm 0.06$ | $0.90 \pm 0.08$ |
| | ST | $\mathbf{0.70} \pm 0.10$ | $\mathbf{0.73} \pm 0.05$ | $\mathbf{0.71} \pm 0.04$ | $\mathbf{0.76} \pm 0.06$ |
| | RF(M=2) T | 37.8 | 43.2 | 68.3 | 69.7 |
| | RF(M=4) T | 31.2 | 41.8 | 56.5 | 63.9 |

**Figure 3:** MNIST classification gradient estimation accuracy and computation cost. *Left, middle:* average (per parameter) squared bias (rsp. variance) in the first layer of the network of different estimators along the training trajectory of ZGR (ternary weights/activations, triangular noise). *Right:* Time of gradient estimate per batch (of size 128) in our implementation. * Our RF implementation is apparently not efficient.

softmax relaxation design. On the other side, we showed that it is unbiased for quadratic functions, generalizing the key property of DARN($\frac{1}{2}$) to the categorical case. We believe that such rationale can be put forward for obtaining improved biased estimators.

On the practical side, ZGR is extremely simple, versatile and computationally inexpensive. In VAE it can replace GR-MC family completely, reducing the computational burden and hyperparameter tuning. It outperforms state-of-the-art unbiased estimators of comparable computation complexity and ST by a large margin. In quantized training it performs close to ST and can fairly replace low-temperature GR-MC variants, while unbiased estimators are completely out of the competition. Thus, across the two corner applications, ZGR is the only estimator which is computationally cheap and well-performing. While in VAE unbiased estimators performs well (and we do not need to worry about the bias) and in quantization simple ST performs well, the above results suggest that there should be cases where ZGR would perform significantly better than both RF and ST. This may be the case for example when considering a different learning formulation such as Bayesian learning of quantized weights or learning with the multi-sample objective Raiko et al. (2015).

## REPRODUCIBILITY STATEMENT

Proofs of all formal claims are presented in Appendix A. Details of the experiments are described in Appendix B. The source code of our implementation will be made publicly available upon publication. During the review period, we will be happy to answer questions and share the code with reviewers confidentially through the OpenReview platform.

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

## A PROOFS

### A.1 BINARY CASE

**Proposition 1.**

$$J_\eta^{\text{GR}} = \frac{\mathcal{L}'(x)}{p(x)} p_Z(\eta) \left( \frac{1}{2} + (2x - 1)c_1 \log(2)t \right) + O(t^2), \tag{10}$$

where $p_Z$ is the logistic density: $p_Z(\eta) = \sigma(\eta)\sigma(-\eta)$ and $c_1 = 2p - 1$.

*Proof.* The conditional density $p(z|x)$ is

$$p(z|x) = \begin{cases} p_Z(z) [\![\eta + z \geq 0]\!]/p(x{=}1), & \text{if } x = 1; \\ p_Z(z) [\![\eta + z < 0]\!]/p(x{=}0), & \text{if } x = 0. \end{cases} \tag{15}$$

Let us denote $p(x{=}1)$ as just $p$. The GR estimator expands as

$$J_\eta^{\text{GR}} = \begin{cases} \frac{\mathcal{L}'(1)}{p} \int_{-\infty}^{\eta} \sigma_t'(\eta + z)p(z)\mathrm{d}z, & \text{if } x = 1; \\ \frac{\mathcal{L}'(0)}{1-p} \int_{\eta}^{\infty} \sigma_t'(\eta + z)p(z)\mathrm{d}z, & \text{if } x = 0. \end{cases} \tag{16}$$

Using the change of variables $v = \sigma_t(\eta + z)$, with the inverse $z = t\text{logit}(v) - \eta$, we have

$$\mathrm{d}v = \sigma_t'(\eta + z)\mathrm{d}z \tag{17}$$

and can write the estimator as

$$J_\eta^{\text{GR}} = \frac{\mathcal{L}'(x)}{p(x)} \left( x \int_{\frac{1}{2}}^{1} p_Z(\eta - t\text{logit}v)\mathrm{d}v + (1 - x) \int_0^{\frac{1}{2}} p_Z(\eta - t\text{logit}v)\mathrm{d}v \right). \tag{18}$$

Note that $p_Z(\eta - t\text{logit}v)$ is bounded above by a constant $\sup_z p_Z(z) = \frac{1}{4}$. A constant is integrable on $[0, 1]$. By dominated convergence theorem we can take the limits $t \to 0$ under the integral. In particular we can use

$$\lim_{t \to 0} p_Z(\eta - t\text{logit}(v)) = \frac{1}{2} p_Z(\eta) \tag{19}$$

under the integral. In order to get a more detailed view, we make the Taylor series expansion of $p_Z(\eta - t\text{logit}(v))$ and substitute it under the integral. With the help of Mathematica (Wolfram Research, 2021) we obtain:

$$J_\eta^{\text{GR}} = \frac{\mathcal{L}'(x)}{p(x)} p_Z(\eta) \left( \frac{1}{2} + (2x - 1)c_1 \log(2)t + c_2 \frac{\pi^2}{6} t^2 \right) + O(t^3), \tag{20}$$

where $c_1 = \tanh(\eta/2) = 2p - 1$ and $c_2 = \frac{1}{2}(1 - 3/(\cosh(\eta) + 1))$. □

**Corollary 1.** In the limit $t \to 0$ the GR estimator becomes the $\text{DARN}(\frac{1}{2})$ estimator.

*Proof.* From the series expansion, the limit $t- \to 0$ is

$$J_\eta^{\text{GR}} = \frac{1}{2} \frac{\mathcal{L}'(x)}{p(x)} p_Z(\eta) = \frac{1}{2} \frac{\mathcal{L}'(x)}{p(x)} p(1 - p), \tag{21}$$

where $p = \sigma(\eta)$. It remains to show that it matches DARN as defined in (4). Note that $\frac{\mathrm{d}\log p(x{=}1;\eta)}{\mathrm{d}\eta} = \sigma(\eta)(1 - \sigma(\eta)) = p(1 - p)$ and $\frac{\mathrm{d}\log p(x{=}0;\eta)}{\mathrm{d}\eta} = -p(1 - p)$. By expanding the cases for $x = 1$ and $x = 0$ we verify that

$$(x - \bar{x}) \frac{\mathrm{d}\log p(x{=}1;\eta)}{\mathrm{d}\eta} = \frac{1}{2} p(1 - p), \tag{22}$$

where $\bar{x} = \frac{1}{2}$. □

**Corollary 2.** The mean and variance of the GR estimator (9) in the asymptote $t \to 0$ are:

$$\mathbb{E}[J_\eta^{\text{GR}}] = p(1 - p) \left( \frac{1}{2}(\mathcal{L}'(1) + \mathcal{L}'(0)) + (\mathcal{L}'(1) - \mathcal{L}'(0))\tilde{c}_1 t \right) + O(t^2), \tag{11a}$$

$$\mathbb{V}[J_\eta^{\text{GR}}] = (p(1 - p))^3 \left( \frac{1}{4}(a - b)^2 + \frac{1}{2}(a^2 - b^2)\tilde{c}_1 t \right) + O(t^2), \tag{11b}$$

where $p = \sigma(\eta)$, $a = \frac{\mathcal{L}'(1)}{p}$, $b = \frac{\mathcal{L}'(0)}{1-p}$ and $\tilde{c}_1 = (2p - 1)\log(2)$.

*Proof.* The mean of the estimator is computed from the series expansion up to the first order as

$$p\frac{\mathcal{L}'(1)}{p}p_Z(\eta)\left(\tfrac{1}{2}+\tilde{c}_1 t\right)+O(t^2)$$

$$+(1-p)\frac{\mathcal{L}'(0)}{1-p}p_Z(\eta)\left(\tfrac{1}{2}+-1\tilde{c}_1 t\right)+O(t^2)$$

$$=p_Z(\eta)\left(\tfrac{1}{2}(\mathcal{L}'(1)+\mathcal{L}'(1))+\tilde{c}_1(\mathcal{L}'(1)-\mathcal{L}'(0))t\right)+O(t^2). \tag{23}$$

Since the GR estimator $J_\eta^{\mathrm{GR}}(x)$ is a Bernoulli variable with values $J_\eta^{\mathrm{GR}}(0)$ and $J_\eta^{\mathrm{GR}}(1)$ with probabilities $p$ and $1-p$, respectively, we can compute is variance simply as

$$\left(J_\eta^{\mathrm{GR}}(1)-J_\eta^{\mathrm{GR}}(0)\right)^2 p(1-p). \tag{24}$$

Using that $p_Z(\eta)=p(1-p)$, the asymptotic expansion of variance up to first order in $t$ is

$$(p(1-p))^3\left(\left(a(\tfrac{1}{2}+\tilde{c}_1 t)-b(\tfrac{1}{2}-\tilde{c}_1 t)\right)\right)^2+O(t^2) \tag{25a}$$

$$=(p(1-p))^3\left(\tfrac{1}{2}(a-b)+(a+b)\tilde{c}_1 t\right)^2+O(t^2), \tag{25b}$$

where $a=\frac{f'(1)}{p}$, $b=\frac{f'(0)}{1-p}$, $\tilde{c}_1=\log(2)c_1$. The first order term is

$$\tfrac{1}{2}(p(1-p))^3(a^2-b^2)\tilde{c}_1 t. \tag{26}$$

It could be positive or negative depending on the values of the derivatives and of $p$. Let us expand $a,b$ and $c_1=\tanh(\eta/2)=2p-1$. We obtain, up to positive constants,

$$p(1-p)(f'(1)^2(1-p)^2-f'(0)^2p^2)(2p-1)t. \tag{27}$$

We see that for the corner points, where $p$ approaches either 0 or 1, this linear term is negative. In particular for a linear objective we have $f'(1)=f'(0)$ and the linear term becomes

$$-p(1-p)f'(1)^2(2p-1)^2 t, \tag{28}$$

which is non-positive for any $p$ and is zero for $p=\tfrac{1}{2}$. $\qquad\square$

## A.2 General Categorical Case

This case is significantly more difficult, as we are dealing with multivariate integration in $K$ Gumbel variables. We will make use of the following statistical relationship.

**Lemma A.1.** *Let $G_1,\dots G_K$ be independent standard Gumbel random variables. Then $Z$ with components $Z_i=G_i-G_K$ for $i=1\dots K-1$ has the multivariate logistic distribution (Malik & Abraham, 1973) with cdf*

$$F_Z(z)=\frac{1}{1+\sum_{i=1}^{K-1}e^{-z_i}}. \tag{29}$$

*Proof.* The cdf and density of Gumbel$(0,1)$ distribution are given respectively by

$$F_G(x)=e^{-e^{-x}};\quad p_G(x)=e^{-(x+e^{-x})}. \tag{30}$$

The conditional distribution of $Z_i$ given $G_K$ is $F_{Z_i|G_K}(z_i|y)=e^{-e^{-(z_i+y)}}$. The conditional joint distribution of $Z$ given $G_K$ is respectively

$$F_{Z|G_K}(z|y)=\prod_{i=1}^{K-1}e^{-e^{-(z_i+y)}}=\exp(\sum_{i=1}^{K-1}-e^{-(z_i+y)})=\exp(-e^{-y}\sum_{i=1}^{K-1}e^{-z_i})=e^{-e^{-y}S}, \tag{31}$$

where $S=\sum_{i=1}^{K-1}e^{-z_i}$. The cdf of $Z$ is obtained by computing the expectation of $F_{Z|G_K}$ in $G_K$:

$$F_Z(z)=\int_{-\infty}^{\infty}e^{-e^{-y}S}e^{-(y+e^{-y})}\mathrm{d}y=\int_{-\infty}^{\infty}e^{-y-e^{-y}(1+S)}\mathrm{d}y$$

$$=\frac{1}{1+S}\int_{-\infty}^{\infty}(1+S)e^{-y-(1+S)e^{-y}}\mathrm{d}y=\frac{1}{1+S}\int_{-\infty}^{\infty}e^{-v-e^{-v}}\mathrm{d}v=\frac{1}{1+S}, \tag{32}$$

where $v=y-\log(S+1)$ and the last equality is by recognizing the Gumbel density under the integral. $\qquad\square$

**Theorem 1** (ZGR). *The Gumbel-Rao estimator for one-hot embedding $\phi$ in the limit of zero temperature is given by*

$$J^{ZGR}_{\theta_i} = \begin{cases} \frac{1}{2}(J_{\phi_i} - J_{\phi_x})p(x{=}i; \eta) & \text{if } i \neq x; \\ -\frac{1}{2}\sum_{j \neq x}(J_{\phi_j} - J_{\phi_x})p(x{=}j; \eta) & \text{if } i = x. \end{cases} \tag{12}$$

*Proof.* We can take $J_\phi = \frac{\mathrm{d}\mathcal{L}(\phi(x))}{\mathrm{d}\phi}$ out of the conditional expectation since it does not depend on $G$ for fixed $x$. Furthermore, $X$ is distributed as $p(x)$ by the sampling procedure and therefore $\mathbb{P}_G(X{=}x) = p(x)$. We can thus rewrite the conditional expectation (8) as

$$J_\phi \int [\![ X(u) = x ]\!]\frac{\mathrm{d}\tilde{\phi}(u)}{\mathrm{d}\theta}\mathrm{d}F_G(u)/\mathbb{P}_G(X{=}x) = \frac{1}{p(x)}J_\phi \int\limits_{\arg\max(\theta+u)=x} \frac{\mathrm{d}\tilde{\phi}(u)}{\mathrm{d}\theta}\mathrm{d}F_G(u), \tag{33}$$

where $F_G$ is the joint cdf of $G$. The condition $\arg\max_k(\theta_k + G_k) = x$ can be expressed as

$$\theta_j + G_j - (\theta_x + G_x) \leq 0, \quad \forall j \neq x. \tag{34}$$

Let us define $\beta_j = \theta_j - \theta_x$ and $Z_j = G_j - G_x$ for $j = 1 \ldots K$. Note that $\beta_x = Z_x = 0$ by this definition. Then the constraint can be written as

$$Z \leq -\beta. \tag{35}$$

The integrand $\tilde{\phi}$ expresses in variables $\beta$, $Z$ as

$$\tilde{\phi} = \mathrm{softmax}_t(\theta + G) = \mathrm{softmax}_t(\beta + Z). \tag{36}$$

Let us denote $Z_{\neg x} = (Z_j | j \neq x)$. The joint distribution of $Z_{\neg x}$ is the $(K-1)$-variate multivariate logistic distribution (Malik & Abraham, 1973), as detailed in Lemma A.1, with cdf:

$$F_{Z_{\neg x}}(z_{\neg x}) = \frac{1}{1+\sum_{i \neq x} e^{-z_i}}. \tag{37}$$

To simplify notation, we let $Z_x$ have the discrete law with mass 1 at a single point $z_x = 0 = -\beta_x$ and extend $F_{Z_{\neg x}}$ to the full joint $F_Z$ accordingly. We then can rewrite the integral as

$$\left(\int\limits_{z \leq -\beta} \frac{\partial}{\partial\beta}\mathrm{softmax}_t(\beta + z)\mathrm{d}F_Z(z)\right)\frac{\partial\beta}{\partial\theta}. \tag{38}$$

The Jacobian $\frac{\partial}{\partial\beta}\mathrm{softmax}_t(\beta + z)$ is a $K \times K$ matrix with indices $(k, j)$ where the column $j = x$ is zero by definition. Let us consider one component of the above integral for $j \neq x$:

$$I_{k,j} = \int\limits_{z \leq -\beta} \frac{\partial}{\partial\beta_j}\mathrm{softmax}_t(\beta + z)_k\mathrm{d}F_Z(z). \tag{39}$$

We want to evaluating its limit for $t \to 0$. We cannot push the limit under the integral in this form, we need to transform it first. To shorten the notation, let us denote $a_i = e^{(z_i + \beta_i)/t}$. We change the variable $z_j$ by the mapping

$$T: z_j \mapsto v_j = (2 + S_j)\frac{a_j}{1 + a_j + S_j}. \tag{40}$$

where $S_j = \sum_{i \neq x,j} a_j$. This mapping is monotone increasing and one-to one from $(-\infty, -\beta)$ to $(0, 1)$, therefore the constraint $z_j \leq -\beta_j$ will trivialize.

Let $A_k = \mathrm{softmax}_t(\beta + z)_k$. We can rewrite the integrand $\frac{\partial}{\partial\beta_j}A_k\mathrm{d}z_j$ as follows:

$$\frac{\mathrm{d}A_k}{\mathrm{d}\beta_j}\mathrm{d}z_j = \frac{\mathrm{d}A_k}{\mathrm{d}z_j}\mathrm{d}z_j \qquad (A_k \text{ depends on } \beta_j \text{ in the same way as on } z_j)$$

$$= \frac{\mathrm{d}A_k}{\mathrm{d}z_j}\frac{\mathrm{d}z_j}{\mathrm{d}v}\mathrm{d}v = \frac{\mathrm{d}A_k}{\mathrm{d}v_j}\mathrm{d}v. \tag{41}$$

We will thus need to evaluate $C_j := \frac{\mathrm{d}A_k}{\mathrm{d}v_j} = \frac{\mathrm{d}A_k}{\mathrm{d}a_j}\frac{\mathrm{d}a_j}{\mathrm{d}v_j} = \frac{\mathrm{d}A_k}{\mathrm{d}a_j}(\frac{\mathrm{d}v_j}{\mathrm{d}a_j})^{-1}$. For $j = k$ we simply have

$$A_k = \frac{a_j}{1 + a_j + S_j} = \frac{1}{2 + S_j}v_j; \quad C_j = \frac{1}{2 + S_j}. \tag{42}$$

For $j \neq k$ we have

$$\frac{\mathrm{d}A_k}{da_j} = \frac{\mathrm{d}}{da_j}\frac{a_k}{1+a_j+S_j} = \frac{-a_k}{(1+a_j+S_j)^2} \tag{43a}$$

$$\frac{dv_j}{da_j} = (2+S_j)\left(\frac{1}{1+a_j+S_j} - \frac{a_j}{(1+a_j+S_j)^2}\right) = \frac{(2+S_j)(1+S_j)}{(1+a_j+S_j)^2} \tag{43b}$$

$$C_j = \frac{-a_k}{(2+S_j)(1+S_j)}. \tag{43c}$$

The integral $I_{k,j}$ with the change of variable $z_j \mapsto v_j$ expresses as

$$\int\limits_{z_i \leq -\beta_i \ \forall i \neq j} \int_0^1 C_j f_t(v_j|z_{\neg j})\mathrm{d}v_j \mathrm{d}F_{Z_{\neg j}}(z_{\neg j}), \tag{44}$$

where $z_{\neg j} = (z_i | i \neq j)$ and

$$f_t(v_j|z_{\neg j}) = p_{Z_j|Z_{\neg j}}(T^{-1}(v_j)|z_{\neg j}). \tag{45}$$

The dependance of $f_t$ on $t$ is through $T$, while $C_j$ depends on $t$ and $z_{\neg j}$. Note that $f$ is a squashed density and is itself not a density.

Next we show dominated convergence of $h_t(v_j, z_{\neg j}) = C_j f_t(v_j|z_{\neg j})$ in $t \to 0$. If $h_t(v_j, z_{\neg j})$ converges point-wise and bounded above by an integrable function, then the limit $t \to 0$ can be taken under the integral.

We show a constant bound on $h_t(v_j, z_{\neg j})$ as follows. Note that $|C_j| \leq 1$. We then have

$$|h_t(v_j, z_{\neg j})| \leq \sup_{v \in (0,1), z_{\neg j}} f(v_j|z_{\neg j}) = \sup_z p_{Z_j|Z_{\neg j}}(z_j|z_{\neg j}), \tag{46}$$

which is the supremum of the conditional density of the standard multivariate distribution and is equal to some constant $c$ independent of $t$. The integral of a constant function $c$ over $(0,1) \times \mathbb{R}^{K-1}$ with respect to the measure $\mathrm{d}v_j \mathrm{d}F_{Z_{\neg j}}(z_{\neg j})$ is $c$.

The point-wise limit is as follows. For $z$ satisfying the constraints $z_i + \beta_i \leq 0$ strictly for $i \neq j, x$, we have

$$\lim_{t \to 0} a_i = \lim_{t \to 0} e^{(\beta_i + z_i)/t} = 0 \ \text{ and } \ \lim_{t \to 0} S_j = 0. \tag{47}$$

Therefore we have

$$\lim_{t \to 0} C_j = \begin{cases} \frac{1}{2} & \text{if } j = k; \\ 0 & \text{if } j \neq k \wedge k \neq x; \\ -\frac{1}{2} & \text{if } j \neq k \wedge k = x. \end{cases} \tag{48}$$

The inverse of mapping $T$ is given by the relations

$$a_j = \frac{v_j(1+S_j)}{2+S_j-v_j}; \quad z_j = -\beta_j + t\log(a_j). \tag{49}$$

It is seen that the limit of $\log(a_j)$ is finite and therefore

$$\lim_{t \to 0} T^{-1}(v_j) = -\beta_j. \tag{50}$$

and

$$\lim_{t \to 0} f_t(v_j|z_{\neg j}) = \lim_{t \to 0} p_{Z_j|Z_{\neg j}}(T^{-1}(v_j)|z_{\neg j}) = p_{Z_j|Z_{\neg j}}(\lim_{t \to 0} T^{-1}(v_j)|z_{\neg j})$$

$$= p_{Z_j|Z_{\neg j}}(-\beta_j|z_{\neg j}). \tag{51}$$

By dominated convergence theorem, we can now claim

$$\lim_{t \to 0} I_{k,j} = 0 \ \text{ if } j \neq k \text{ and } k \neq x. \tag{52}$$

And elsewise, if $j = k$ or $k = x$,

$$\lim_{t \to 0} I_{k,j} = \int\limits_{z_i \leq -\beta_i \ \forall i \neq j} \pm \frac{1}{2} p_{Z_j|Z_{\neg j}}(-\beta_j|z_{\neg j})\mathrm{d}F_{Z_{\neg j}}(z_{\neg j}) = \pm\frac{1}{2}\frac{\partial}{\partial z_j}F_Z(z)\Big|_{z=-\beta} \tag{53}$$

$$= \mp\frac{1}{2}\frac{\partial}{\partial \beta_j}\frac{1}{1+\sum_{i \neq x}e^{\beta_i}} = \mp\frac{1}{2}\frac{\partial}{\partial \beta_j}\frac{e^{\beta x}}{\sum_i e^{\beta_i}} = \mp\frac{1}{2}\frac{\partial}{\partial \beta_j}\text{softmax}(\beta)_x \tag{54}$$

$$= \pm\frac{1}{2}p(x)p(j), \tag{55}$$

where the upper sign corresponds to the case $j = k$ and the lower to $k = x$.

Let us denote $\hat{I} = \lim_{t \to 0} I$. Multiplying it with the incoming derivative $J_\phi$ on the left, we obtain:

$$(J_\phi \hat{I})_j = \tfrac{1}{2}(J_{\phi_j} - J_{\phi_x})p(x)p(j). \tag{56}$$

And finally, multiplying (56) with the Jacobian $\frac{\partial \beta}{\partial \theta}$ on the right per (38) and with the factor $\frac{1}{p(x)}$ per (33), we obtain

$$J_{\theta_i}^{\text{ZGR}} = \begin{cases} \tfrac{1}{2}(J_{\phi_i} - J_{\phi_x})p(i) & \text{if } i \neq x; \\ -\tfrac{1}{2}\sum_{j \neq x}(J_{\phi_j} - J_{\phi_x})p(j) & \text{if } i = x. \end{cases} \tag{57}$$

$\square$

**Proposition 2.** ZGR estimator decomposes as

$$\boxed{J^{\text{ZGR}} = \tfrac{1}{2}(J^{\text{ST}} + J^{\text{DARN}(\bar{\phi}(\eta))}),} \tag{13}$$

*i.e.*, with the choice $\bar{\phi} = \bar{\phi}(\eta)$ in DARN.

*Proof.* Let $p$ denote the vector of probabilities $(p(x{=}k; \eta)|k = 0, \dots, K - 1)$. Recall that we have derived ZGR under the assumption of on-hot embedding $\phi$, inherited from GS. In this case $J_\phi \phi(i) = J_{\phi_i}$ and $\bar{\phi}_k = \sum_i \phi(i)_k p_i = p_k$.

Note that ZGR (57) defines the gradient in the parametrization $\theta$ used in Gumbel Rao and initially in Gumbel-Softmax, while ST and DARN estimators are given by us with respect to $\eta$. We need to bring these two to a common basis. We chose to reconstruct $J_p^{\text{ZGR}}$ because both $J_p^{\text{ST}}$ and $J_p^{\text{DARN}}$ are particularly simple:

$$J_{p_i}^{\text{ST}} = J_\phi \phi(i) = J_{\phi_i}, \tag{58}$$
$$J_{p_i}^{\text{DARN}} = J_\phi(\phi_i - \bar{\phi})[\![x{=}i]\!]/p(x) = (J_{\phi_i} - J_\phi p)[\![x{=}i]\!]/p(x). \tag{59}$$

Note, because $p$ lies in the simplex, gradients in $p$ are defined up to an additive constant to all coordinates. In other words any such additive constant is irrelevant and will not affect the gradient in $\eta$.

In order to reconstruct $J_p^{\text{ZGR}}$ we represent $J_\theta^{\text{ZGR}} = J_p^{\text{ZGR}}P$, where $P$ is the Jacobian of softmax, given by

$$P = \text{diag}(p) - pp^\mathsf{T} = \text{diag}(p)(I - \mathbf{1}p^\mathsf{T}). \tag{60}$$

We first note that $J_\theta^{\text{ZGR}}$ satisfies $\sum_i J_{\theta_i}^{\text{ZGR}} = 0$ (as any gradient should, but not necessarily a stochastic estimator) and therefore

$$J_\theta^{\text{ZGR}} = J_\theta^{\text{ZGR}}(I - \mathbf{1}p^\mathsf{T}) = J_\theta^{\text{ZGR}}\text{diag}(p)^{-1}P. \tag{61}$$

We obtained:

$$J_{p_i}^{\text{ZGR}} = \begin{cases} \tfrac{1}{2}(J_{\phi_i} - J_{\phi_x}) & \text{if } i \neq x; \\ -\tfrac{1}{2}\sum_{j \neq x}(J_{\phi_j} - J_{\phi_x})p(j)/p(x) & \text{if } i = x, \end{cases} \tag{62}$$

up to a constant, *i.e.* adding the same number $c$ to all components. We further add the constant $\tfrac{1}{2}J_{\phi_x}$ and obtain

$$J_{p_i}^{\text{ZGR}} = \begin{cases} \tfrac{1}{2}J_{\phi_i} & \text{if } i \neq x; \\ \tfrac{1}{2}J_{\phi_x} - \tfrac{1}{2}\sum_{j \neq x}(J_{\phi_j} - J_{\phi_x})p(j)/p(x) & \text{if } i = x, \end{cases} \tag{63}$$

Subtracting $\tfrac{1}{2}J_p^{\text{ST}}$, the reminder is $\tfrac{1}{2}J_p^{\text{RE}}$ with

$$J_{p_i}^{\text{RE}} = [\![i{=}x]\!]\tfrac{1}{p(x)}\sum_{j \neq x}(J_{\phi_x} - J_{\phi_j})p(j). \tag{64}$$

Simplifying

$$\sum_{j \neq x}(J_{\phi_x} - J_{\phi_j})p(j) = \sum_j(J_{\phi_x} - J_{\phi_j})p(j) = J_{\phi_x} - \sum_j J_{\phi_j}p(j) \tag{65}$$

we obtain

$$J_{p_i}^{\text{RE}} = [\![i{=}x]\!]\tfrac{1}{p(x)}\left(J_{\phi_x} - \sum_j J_{\phi_j}p(j)\right). \tag{66}$$

and we see that $J_p^{\text{RE}} = J_p^{\text{DARN}}$ with $\bar{\phi} = p = \bar{\phi}(\eta)$. $\square$

**Theorem 2.** *ZGR is unbiased for quadratic loss functions $\mathcal{L}$ with arbitrary coefficients.*

*Proof.* Since ZGR estimator is linear in $\mathcal{L}$ (estimate for a linear combination of two loss functions is the linear combination of estimates), it is sufficient to prove the claim for one-hot embedding $\phi$ and some elementary functions forming a basis for all quadratic functions. With one-hot embedding we have $\bar{\phi}(\eta)_i = p(x{=}i; \eta) = p_i$.

Let us start with a linear monomial $\mathcal{L}(x) = \phi(x)_i$. The expected loss is $\mathbb{E}[\mathcal{L}(x)] = p(x{=}i; \eta)$. The true gradient is

$$J_\eta = \tfrac{\mathrm{d}}{\mathrm{d}\eta} p(x{=}i; \eta). \tag{67}$$

Substituting $J_{\phi_k} = [\![k{=}i]\!]$ in ST we have

$$J_\eta^{\mathrm{ST}} = \tfrac{\mathrm{d}\mathcal{L}(\phi(x))}{\mathrm{d}\phi} \tfrac{\mathrm{d}\bar{\phi}(\eta)}{\mathrm{d}\eta} = \tfrac{\mathrm{d}\bar{\phi}(\eta)_i}{\mathrm{d}\eta} = J_\eta. \tag{68}$$

This may come as a surprise for someone, but ST for a single categorical variable is exact (zero bias and zero variance). The expectation of $J^{\mathrm{DARN}}$ simplifies as follows for any $\bar{\phi}$ and a linear loss function, ensuring that $J_\phi$ is constant in $x$:

$$\mathbb{E}[J_\eta^{\mathrm{DARN}}] = \sum_x p(x) J_\phi (\phi(x) - \bar{\phi}) \tfrac{1}{p(x)} \tfrac{\mathrm{d}p(x;\eta)}{\mathrm{d}\eta} = J_\phi \sum_x (\phi(x) - \bar{\phi}) \tfrac{\mathrm{d}p(x;\eta)}{\mathrm{d}\eta}$$
$$= J_\phi \sum_x \phi(x) \tfrac{\mathrm{d}p(x;\eta)}{\mathrm{d}\eta} - J_\phi \bar{\phi} \tfrac{\mathrm{d}}{\mathrm{d}\eta} \sum_x p(x;\eta) = J_\phi \tfrac{\mathrm{d}\bar{\phi}(\eta)}{\mathrm{d}\eta}. \tag{69}$$

Substituting $J_{\phi_k} = [\![k{=}i]\!]$ and $\bar{\phi}(\eta) = p$ we obtain

$$\mathbb{E}[J_\eta^{\mathrm{DARN}}] = \tfrac{\mathrm{d}p(x{=}i;\eta)}{\mathrm{d}\eta} = J_\eta, \tag{70}$$

reconfirming that DARN is unbiased for linear function of categorical variables as expected. It follows that $\frac{1}{2}(J_\eta^{\mathrm{ST}} + J_\eta^{\mathrm{DARN}})$ is also unbiased.

Let us now consider the elementary quadratic function $\mathcal{L}(\phi(x)) = \phi(x)_i^2 - \phi(x)_i$. For all discrete assignments it is zero, therefore the true gradient of its expected value is zero. We have

$$J_{\phi_k}(x) = \begin{cases} 2\phi(x)_i - 1 & k = i \\ 0 & k \neq i. \end{cases} \tag{71}$$

Therefore $J_{p_k}^{\mathrm{ST}} = 0$ for $k \neq i$ and

$$\mathbb{E}[J_{p_i}^{\mathrm{ST}}] = \mathbb{E}[2\phi(x)_i - 1] = 2p_i - 1. \tag{72}$$

For $J_{p_i}^{\mathrm{DARN}}$ we have

$$J_{p_i}^{\mathrm{DARN}} = (J_{\phi_i}(x) - J_\phi(x)p) \tfrac{1}{p(x)} [\![x{=}i]\!] \tag{73}$$
$$= (1 - p_i)(2\phi(x)_i - 1) \tfrac{1}{p(x)} [\![x{=}i]\!]. \tag{74}$$

Its expectation is

$$(1 - p_i)(2\phi_i(i) - 1) = 1 - p_i. \tag{75}$$

For $J_{p_k}^{\mathrm{DARN}} = 0$ for $k \neq i$ we have

$$J_{p_k}^{\mathrm{DARN}} = (J_{\phi_k}(x) - J_\phi(x)p) \tfrac{1}{p(x)} [\![x{=}k]\!] \tag{76a}$$
$$= -J_\phi(x)p \tfrac{1}{p(x)} [\![x{=}k]\!] \tag{76b}$$
$$= -p_i(2\phi(x)_i - 1) \tfrac{1}{p(x)} [\![x{=}k]\!]. \tag{76c}$$

Its expectation is

$$-p_i(2\phi_i(k) - 1) = p_i. \tag{77}$$

By subtracting $p_i$ from all ordinates of $J_p^{\mathrm{DARN}}$, we obtain an equivalent (having identical derivative in $\eta$) form where $\mathbb{E}[J_{p_k}^{\mathrm{DARN}}] = 0$ for all $k \neq i$ and $\mathbb{E}[J_{p_i}^{\mathrm{DARN}}] = 1 - 2p_i$, which cancels with $\mathbb{E}[J_{p_i}^{\mathrm{ST}}]$.

Next we consider a bilinear monomial in $\phi$: $\mathcal{L}(\phi(x)) = \phi(x)_1\phi(x)_2$, where we have taken indices 1 and 2, without loss of generality. Its is zero for all discrete assignments and therefore the gradient of its expectation is zero. We have

$$J_{phi_1} = \phi_2(x) = [\![x{=}2]\!] \tag{78a}$$

$$J_{phi_2} = \phi_1(x) = [\![x{=}1]\!]. \tag{78b}$$

For ST we have $J_p^{\mathrm{ST}} = J_\phi$ and

$$\mathbb{E}[J_{p_1}^{\mathrm{ST}}] = p_2, \tag{79a}$$

$$\mathbb{E}[J_{p_2}^{\mathrm{ST}}] = p_1, \tag{79b}$$

$$\mathbb{E}[J_{p_k}^{\mathrm{ST}}] = 0, \quad k \neq 1, 2. \tag{79c}$$

For DARN part we have:

$$J_{p_1}^{\mathrm{DARN}} = [\![x{=}1]\!]\tfrac{1}{p_1}(J_{\phi_1} - J_{\phi_1}p_1 - J_{\phi_2}p_2), \tag{80a}$$

$$J_{p_2}^{\mathrm{DARN}} = [\![x{=}2]\!]\tfrac{1}{p_2}(J_{\phi_2} - J_{\phi_1}p_1 - J_{\phi_2}p_2), \tag{80b}$$

$$J_{p_k}^{\mathrm{DARN}} = [\![x{=}k]\!]\tfrac{1}{p_k}(-J_{\phi_1}p_1 - J_{\phi_2}p_2), \quad k \neq 1, 2. \tag{80c}$$

In the expectation, substituting $J_\phi$:

$$\mathbb{E}[J_{p_1}^{\mathrm{DARN}}] = [\![x{=}1]\!]\tfrac{1}{p_1}(\phi_2(1) - \phi_2(1)p_1 - \phi_1(1)p_2) = -p_2, \tag{81a}$$

$$\mathbb{E}[J_{p_2}^{\mathrm{DARN}}] = [\![x{=}2]\!]\tfrac{1}{p_2}(\phi_1(2) - \phi_2(2)p_1 - \phi_1(2)p_2) = -p_1, \tag{81b}$$

$$\mathbb{E}[J_{p_k}^{\mathrm{DARN}}] = [\![x{=}k]\!]\tfrac{1}{p_k}(-\phi_2(k)p_1 - \phi_1(k)p_2) = 0, \quad k \neq 1, 2. \tag{81c}$$

This exactly cancels with ST.

The elementary functions we have considered form a basis in the space of all quadratic functions. By linearity argument, $J^{ZGR} = \frac{1}{2}(J^{ST} + J^{DARN})$ is unbiased for all quadratic functions. $\qquad\square$

# B  Details of Experiments

Here we give detailed specifications of our experiments. The implementation of all experiments will be made publicly available upon publication. During the review period, we will be happy to answer questions and share the code with reviewers confidentially through the OpenReview platform.

## B.1  Dataset

In quantized training we use **MNIST**[1] and **FashionMNIST**[2] datasets. Each contains 60000 training and 10000 test images. We used 54000 images for training and 6000 for validation.

In VAE training, following the prior work, we use a decoder with Bernoulli output layer, which requires binary datasets. **MNIST-B** is a binarized MNIST with a fixed threshold of 0.5, same as in Yin et al. (2019). The original Omniglot dataset is of the size $105 \times 105$ and contains binary images. However the established benchmarks use its down-sampled version (to size $28 \times 28$), which is then dynamically sampled: binary pixel values are generated with probabilities proportional to the original pixel values (Burda et al., 2016; Dong et al., 2021), which we denote as **Omniglot-28-D**. The down-scaled dataset published by Burda et al. (2016)[3] was used, same as in the public implementation of Dong et al. (2021). It contains about 24000 training images, which were split into training (90%) and validation (10%) parts and currently we are not using the validation part.

---

[1] http://yann.lecun.com/exdb/mnist/
[2] https://github.com/zalandoresearch/fashion-mnist
[3] https://github.com/yburda/iwae/raw/master/datasets/OMNIGLOT/chardata.
mat

```
def ZGR(p:Tensor)->Tensor:
    """Returns a categorical sample from p [*,C] (over axis=-1) as
        one-hot vector, with ZGR gradient.
    """
    index = Categorical(probs=p).sample()
    x = F.one_hot(index, num_classes=p.shape[-1]).to(p)
    logpx = p.log().gather(-1, index.unsqueeze(-1))# log p(x)
    dx_ST = p
    dx_RE = (x - p.detach()) * logpx
    dx = (dx_ST + dx_RE) / 2
    return x + (dx - dx.detach()) # value of x with backprop through dx
```

**Figure B.1:** ZGR implementation in Pytorch for a general categorical variable.

## B.2    METHODS

**ZGR** for categorical variables can be implemented as shown in Fig. B.1. It is a plug-in estimator, meaning that it is sufficient to use it for every tensor of categorical variables in a hierarchical model and the gradient in all parameters will be computed by back-propagation automatically.

Gumbel-Softmax (**GS**) and Straight-through Gumbel-Softmax (**GS-ST**) (Jang et al., 2017) are shipped with pytorch[4].

For Gumbel-Rao with MC samples (**GR**) we adopted the public reimplementation by nshepperd [5].

**RF**($M$) we implemented according to (Kool et al., 2019, Eq. 8). The part of the computation relevant to the encoder is propagated forward and backward only once. In the decoder we perform as many backward passes as forward, as this reduces variance of the gradient in decoder parameters. In quantization our implementation performs a backward pass for each forward pass, and is not well organized.

For **ARSM** (Yin et al., 2019) we made own reimplementation, cross-checked with the authors tensor-flow implementation [6]. As with RF($M$), we also performed a backward pass for each forward pass.

## B.3    VAE

**Model**    In our model each categorical variable is encoded as a vector of $\pm 1$, corresponding to the bit representation of $x$, similar to Paulus et al. (2021). There is a fixed number of total hidden bits (192), which are split into several categorical variables. For example 192 1b variables or 32 6-bit variables. This way the number of weights in the network stays constant. The network architecture is adopted from Yin et al. (2019):

$$\text{Linear(784,512)} \to \text{LReLU} \to \text{Linear(512,256)} \to \text{LReLU} \to \text{Linear(256, D*K)},$$

where in the last layer we have $D$ of $K$-way categorical units and LReLU has a leaky coefficient of 0.2 (same as in Dong et al. (2021), default in tensorflow). The output of the encoder defines logits of the encoder Bernoulli model $q(z_i{=}1|x)$, where $x$ is the input binary image and $z$ is the latent discrete state.

The decoder has exactly the reverse Linear-LReLU architecture and outputs logits of conditionally independent Bernoulli generative model $p(x_i{=}1|z)$. We optimize the standard evidence lower bound (ELBO) Kingma & Welling (2013) with prior distribution $p(z)$ uniform and not learned.

We do not perform any special data-based initializations like subtracting data mean in the encoder in Dong et al. (2021).

**Optimization**    In the forward pass all methods produce a sample, from which a stochastic esti-mate of the gradient with respect to the decoder parameters is readily computed by backpropagation through decoder. We compute the KL term in ELBO analytically for a mini-batch and use its exact

---

[4]Pytorch function `torch.functional.gumbel_softmax`
[5]https://github.com/nshepperd/gumbel-rao-pytorch
[6]https://github.com/ARM-gradient/ARSM

gradient. The estimation problem (1) occurs for the gradient of the data term with respect to the encoder parameters, where the estimators through discrete variables are applied.

All methods, including GS that optimizes ELBO with relaxed samples, are evaluated by the correct ELBO with discrete samples.

In the VAE experiments we measure the gradient accuracy and the training performance and do not make use of validation or test sets. First, this is reasonable when comparing quality of gradient estimators, regardless generalization. Second, the prior work Dong et al. (2021) has verified that improvement in the training ELBO translates into improvement of the test ELBO and IWAE bounds.

Following Dong et al. (2021) we train with Adam with learning rate $10^{-4}$ using batch size 50. Furthermore we tried to match the training time that of Dong et al. (2021). For MNIST we perform 500 epochs, and for Omniglot-28-D we perform 1000 epochs, roughly equivalent in booth cases to their $500K$ iterations with batch size 50.

## B.4 ADDITIONAL VAE EXPERIMENTS

We include some extended results.

**Bias-variance analysis** We conducted the bias-variance analysis for VAE at different training stages. Namely, we trained the model using RF(4) for 1, 10, 100, and 200 epochs and at each stage evaluated bias and variance of all gradient estimators. The model used 16 categorical variables of 16 categories (64 total latent bits). The results are displayed in Fig. B.2. At the very beginning of training, the picture looks substantially different in that there is some bias reversal in GS-ST and derived estimators. However from epoch 10 and on the trends and relative ordering of methods stabilizes, with only RF(4) slightly overtaking ZGR in variance. The different picture after 1 epoch suggests that it would be beneficial in practice to warm-up the training with a few epochs of GS-ST at $t = 1$ or ST. We left such tuning to future work.

## B.5 QUANTIZED NEURAL NETWORKS

**Experimental Setup** In this experiment we train a convolutional network 32C5-MP2-64C5-MP2-512FC-10FC, closely replicating the model evaluated by Louizos et al. (2019) for MNIST. Each activation quantization is preceded by batch normalization (Ioffe & Szegedy, 2015).

All gradient estimators are working with the same network, parametrization and initialization. In the case of logistic noise the noise standard deviation is learnable and is initialized to $1/3$. All methods are applied with Adam optimizer for 200 epochs. For every method we select the bast validation performance with the grid search for the learning rate from $\{10^{-3}, 3.3 * 10^{-4}, 10^{-4}\}$. We used the step-wise learning rate schedule decreasing the learning rate 10 times at epochs 100 and 150. The whole procedure is repeated for 3 different initialization seeds and we report the mean test error over seeds and $\pm(\max - \min)/2$ over seeds.

For validation and testing, we evaluate the network in the 'deterministic' mode, turning off all injected noises. This corresponds to a simple deterministic quantized model to be deployed.

**Single Unit Quantization** We include the following toy experiment that well illustrates properties of different estimators. We evaluate bias and variance of all estimators on a simple function of a single quantized variable. Let $\eta$ be a real-valued parameter. Let $p(x; \eta)$ be given by the stochastic quantization model with $K = 4$ states and a particular noise type. Given a test function $\mathcal{L}(x)$ we can compute the true gradient of $\mathbb{E}[\mathcal{L}(x)]$. For each estimator we draw $10^4$ samples to compute its mean and standard deviation for each value of $\eta$. The results are presented in Fig. B.3. In this plot we show several combinations of loss functions and noises. The test functions are: *linear* $\mathcal{L}(x) = x$; *quadratic* $\mathcal{L}(x) = \frac{1}{2}(x - c)^2$ and *sigmoid* $\mathcal{L}(x) = \sigma(2(x - c))$, where $c = (K - 1)/2$ is chosen for centering. The noises shown refer to the *logistic* noise with std = 1/3 as used at initialization by Louizos et al. 2019 and the *triangular* noise with the density $p(z) = \max(0, 1 - |z|)$.

The bias of the GS family quickly decreases with the temperature. ZRG estimator achieves the same expected value as GS-ST in the limit of small temperature illustrated by GS-ST(t=0.1) and the variance comparable to that of GR(t=0.1, M=100). We also verify that ZGR has zero bias for quadratic objectives as we have shown theoretically.

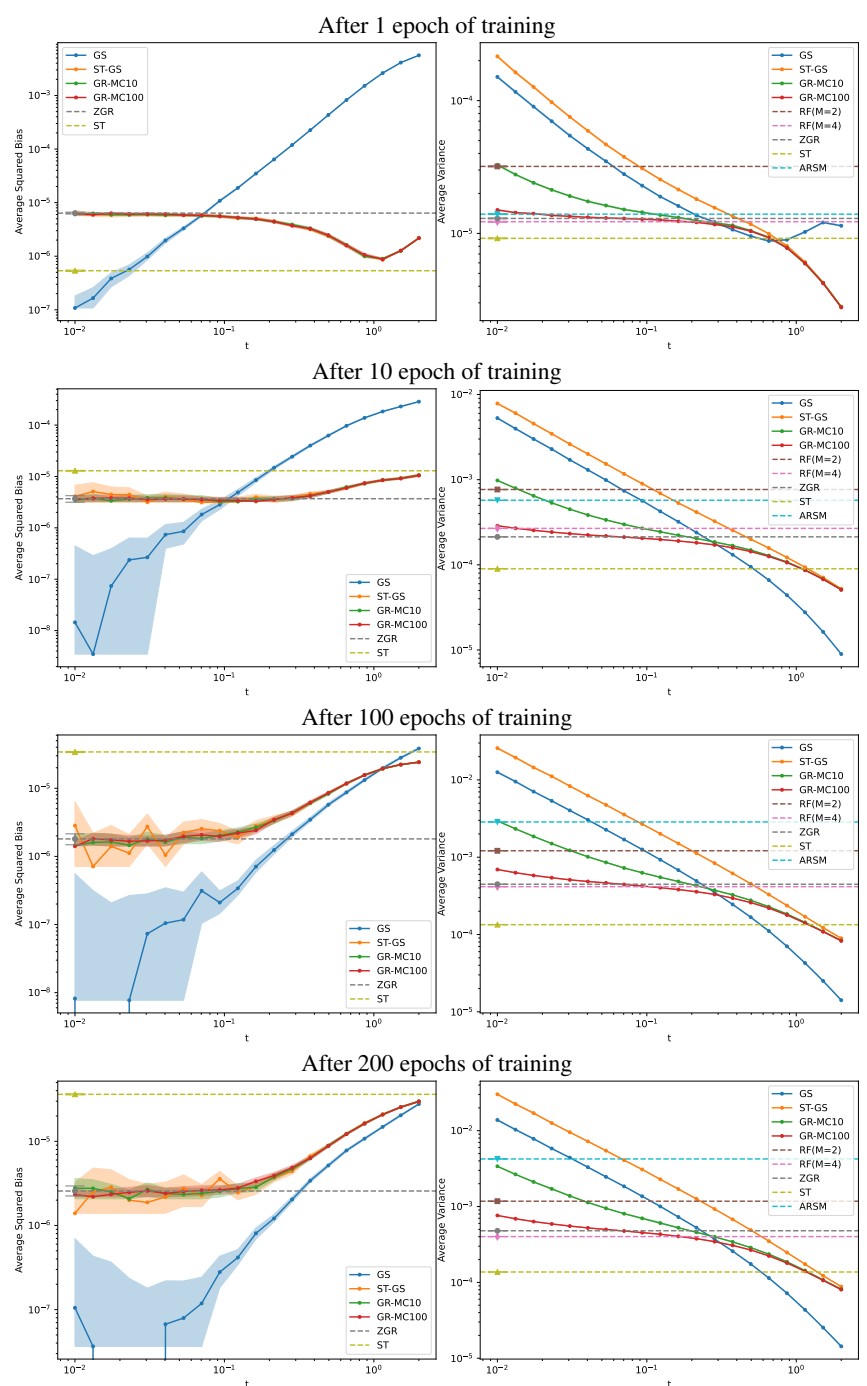

**Figure B.2:** Gradient estimation accuracy in VAE on MNIST-B. Average (per parameter) squared bias (*left*) and variance (*right*) of gradient estimators versus temperature for a model snapshot at a particular iteration of training with RF(4). VAE network with 16 categorical variables with 16 categories.

**Additional Comparisons** In Table 3 we left out GS method after preliminary testing. These results (without confidence bounds) are available in Table B.1. A we also performed a full comparison on FMNIST dataset in Table B.2, which qualitatively agrees with the results for MNIST Table 3 and confirms that ZGR improves over GR-MC(t=0.1, M=10).

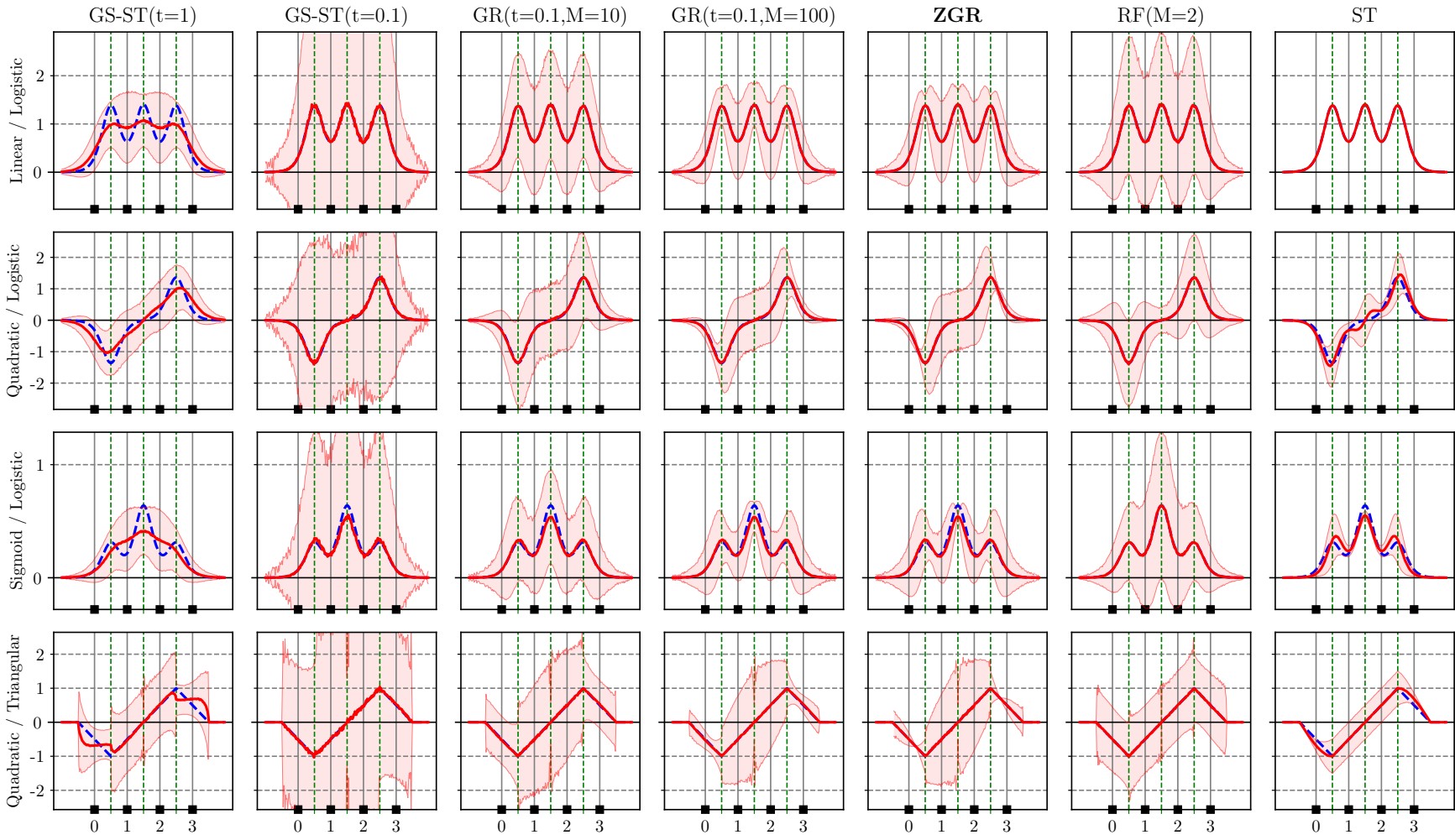

**Figure B.3:** Performance of estimators in stochastic quantization with selected combinations of test functions and injected noise for varied input $\eta$ of the stochastic quantizer. The dashed blue line is the exact gradient. Red line is the mean of the estimator and the red shaded area shows $\pm 1$ std. GR with 100 is able to reduce the variance of GS-ST substantially. ZGR reduces the variance by an edge further while keeping the bias equal to the theoretical bias of GR with zero temperature. The variance of ARSM is substantially higher. Plain ST estimator has a yet smaller variance but a larger bias, which may accumulate during taring.

**Table B.1:** MNIST test error[%] complementing Table 3 with GS variants, which are found to be inferior.

| | **Logistic Noise Relaxation** | | | |
| --- | --- | --- | --- | --- |
| | Weights [bits] / Activations [bits] | | | |
| method | 2/2 | T/T | T/1 | 1/1 |
| GS(t=2) | 0.80 | 0.78 | 0.87 | 1.11 |
| GS(t=1) | 0.86 | 0.72 | 0.98 | 0.86 |
| GS(t=0.5) | 0.94 | 0.82 | 0.99 | 1.18 |

**Table B.2:** FMNIST test error[%] in deterministic mode (no injected noises at test time) for different bit-width per weigh and activation (T denotes ternary). Hyperparameters are selected on the validation set. Reference test errors: ReLU 9.73% , Clamp 9.6%.

| | | Weights [bits] / Activations [bits] | | | |
| --- | --- | --- | --- | --- | --- |
| | Method | 2/2 | T/T | T/1 | 1/1 |
| **Logistic Noise** | GS-ST(t=2) | $8.15 \pm 0.30$ | $8.32 \pm 0.13$ | $8.58 \pm 0.18$ | $8.92 \pm 0.08$ |
| | GS-ST(t=1) | $8.56 \pm 0.23$ | $8.59 \pm 0.30$ | $8.99 \pm 0.22$ | $9.07 \pm 0.05$ |
| | GS-ST(t=0.5) | $9.11 \pm 0.32$ | $9.60 \pm 0.19$ | $9.85 \pm 0.04$ | $10.99 \pm 0.26$ |
| | GR-MC(t=0.5,M=10) | $9.04 \pm 0.23$ | $8.84 \pm 0.10$ | $9.41 \pm 0.10$ | $9.97 \pm 0.08$ |
| | GR-MC(t=0.1,M=10) | $9.35 \pm 0.12$ | $10.10 \pm 0.19$ | $10.63 \pm 0.16$ | $11.61 \pm 0.35$ |
| | ZGR | $9.12 \pm 0.33$ | $9.33 \pm 0.22$ | $9.85 \pm 0.25$ | $10.67 \pm 0.13$ |
| | ST | $8.55 \pm 0.11$ | $8.79 \pm 0.10$ | $9.62 \pm 0.09$ | $9.28 \pm 0.17$ |
| **Triangular Noise** | GS-ST(t=2) | $8.57 \pm 0.13$ | $8.76 \pm 0.14$ | $10.19 \pm 0.09$ | $10.92 \pm 0.30$ |
| | GS-ST(t=1) | $8.59 \pm 0.01$ | $8.94 \pm 0.21$ | $9.95 \pm 0.09$ | $10.56 \pm 0.26$ |
| | GS-ST(t=0.5) | $9.35 \pm 0.29$ | $9.63 \pm 0.20$ | $10.70 \pm 0.13$ | $11.34 \pm 0.07$ |
| | GR-MC(t=0.5,M=10) | $8.61 \pm 0.10$ | $8.62 \pm 0.07$ | $9.34 \pm 0.10$ | $9.76 \pm 0.21$ |
| | GR-MC(t=0.1,M=10) | $8.85 \pm 0.13$ | $9.32 \pm 0.04$ | $10.16 \pm 0.06$ | $10.83 \pm 0.19$ |
| | ZGR | $8.67 \pm 0.25$ | $8.96 \pm 0.25$ | $9.73 \pm 0.19$ | $10.09 \pm 0.38$ |
| | ST | $8.59 \pm 0.20$ | $8.61 \pm 0.17$ | $9.15 \pm 0.18$ | $9.98 \pm 0.19$ |

