# OpenReview forum: "Cold Rao-Blackwellized Straight-Through Gumbel-Softmax Gradient Estimator"
_ICLR.cc/2023/Conference — Submitted to ICLR 2023_

### Official Review · Reviewer_GHDv · 2022-10-24

**Confidence:** 3
**Correctness:** 3
**Technical Novelty And Significance:** 3
**Empirical Novelty And Significance:** 3
**Recommendation:** 3

**Clarity, Quality, Novelty And Reproducibility:**

This paper is neatly written and theoretically qualitative, and therefore has originality.

However, the source code is not available at the moment.

Here are some questions to the authors.

- What do you think complemented ST and DARN, each other?

- In Figure 1, does temperature annealing utilized in the baselines? Or, is it just a figure from bunch of fixed temperatures?

- Same question on Figure 2. Does GS variants utilized the temperature annealing? If not, I think is unfair not to utilize the temperature annealing since it is crucial hyper-parameter is the GS variants.

- Perhaps, there is an inconsistency of temperature variable.


**Strength And Weaknesses:**

The paper is theoretically grounded and the proposed ZGR surprisingly filling the gap between ST and DARN.

Error bars are missing in some tables and figures. Also, the authors did not explore various temperatures (and temperature scheduling) for the baselines, which makes unfair comparison. Finally, the experimental result in Table 1 is not very promising.


**Summary Of The Paper:**

This paper proposes Zeroed Gumbel-Rao (ZGR) stochastic gradient estimators for the discrete random variables. The authors prove that the limiting the temperature to zero in the Gumbel-Rao results in closed-from solution. They also show that the proposed ZGR estimator is in the middle of two well-known stochastic gradient estimators, namely Straight-Through estimator and DARN estimator. They argue that the proposed ZGR has merits both the low bias & variance perspective (when the temperature is less than 0.1), and the low computational complexity, based on their experimental results.


**Summary Of The Review:**

I am very neutral on this paper. The work is theoretic and seems to be correct. However, the concern on the fair comparison in the experiment section needs to be resolved, since the GS variants are sensitive to the temperature tuning.

---

> ### Author Response · Authors · 2022-11-16
> **Discussion**
>
> Thanks for your peer review. We would appreciate further discussion on some points you mentioned.
>
> > What do you think complemented ST and DARN, each other?
>
> From our theory we have that both ST and DARN are unbiased for linear functions (proof of Theorem 2). For quadratic functions both of them do not seem to have a rationale in their design. In particular DARN was extended to categorical variables by Gu et al. (2016) in an purely heuristic fashion. As we mention in the paper, there is no such $\bar \phi$ that DARN($\bar \phi$) would be unbiased for quadratic functions. According to our results ST and DARN($\phi(\eta)$) happen to cancel their biases for quadratic functions.
>
> We would like to further clarify how this is compatible with the results for the binary case, where it is known that DARN(1/2) is unbiased for quadratic functions by design (Gregor et al. 2014). We will clarify in the paper that in the categorical case (Proposition 2) we refer to DARN with $\bar\phi = \phi(\eta)$. Then it is easy to verify that (ST+DARN($\phi(\eta)$))/2 is exactly DARN(1/2) in the binary case. Thus the result we obtained for binary case (Corollary 1) is fully consistent with the general categorical case (Proposition 2).
>
> > In Figure 1, does temperature annealing utilized in the baselines? Or, is it just a figure from bunch of fixed temperatures?
>
> Figure 1 studies bias and variance of temperatured estimators at a given model snapshot (after 100 training iterations with ARSM).
> There is ni training happening in this figure, the annealing is not relevant.
>
> > Same question on Figure 2. Does GS variants utilized the temperature annealing? If not, I think is unfair not to utilize the temperature annealing since it is crucial hyper-parameter is the GS variants.
>
> We can address the temperature hyperparameter to some extent. We conduct experiments in Fig 2 (to be extended in the revision) using only t=0.1 (which we somehow implied from Paulus et a. 2021). We made a sweep over temperatures training for 500 epochs with that temperature to demonstrate that temperature 0.1 is amongst the optimal ones. At the same time we find that such hyperparameter optimization for the baselines puts our method at a disadvantage. Should we also introduce some hyperparameter (perhaps artificially) in ZGR in order to benefit from optimizing it in such comparison?
>
> Coming back to your original question, which was about **temperature annealing**. Clearly, an annealing requires some schedule , which introduces more hyperparameters to the baseline. It becomes computationally infeasible for us to reliably search over them and it gives even more advantage to the baseline. We therefore won't be able to address the temperature annealing question in the rebuttal, but we will appreciate your further thoughts on how one should do it.
>
> We haven't seen a comparison using temperature annealing in the prior work and in fact we fix some other shortcomings with the prior comparisons. For example: Yin et al. 2019 treat $t$ as a learnable parameter, which results in a poor performance.  Furthermore, Paulus et al. 2021 make 50000 steps with batch 20 amounting to only 20 epochs and Yin et al. 2019 by similar calculations perform 150 epochs. The ordering of methods may change with more epochas as visible e.g. in Fig. 1 of Dong et al. "Coupled Gradient Estimators for Discrete Latent Variables".
>
> > Error bars are missing in some tables and figures,  inconsistency of temperature variable.
>
> Will be fixed in the revision. We are preparing extended experiments which include more datasets and error bars from several runs of each method.

---

> > ### Comment · Reviewer_GHDv · 2022-12-12
> > **Update review**
> >
> > As the authors mentioned, temperature requires some scheduling.
> > However, the temperature scheduling is a crucial hyper-parameter tuning for the baselines, which significantly impacts on the experimental result.
> > Meanwhile, the authors did not provide proper reasons for skipping this temperature annealing.
> > I do recognize that the prior works did not compare together with the temperature scheduling, but the following rebuttal from the authors did not convince me.
> > > We haven't seen a comparison using temperature annealing in the prior work ...
> >
> > Hence, my concern is not fully addressed, and I decided to decrease my score to 3.

---

> > > ### Author Response · Authors · 2022-12-12
> > > **temperature scheduling**
> > >
> > > We agree that temperature scheduling might be beneficial in practice. Even if so, it would follow from our VAE evaluation that if the temperature schedule is used with GS-ST it would be beneficial in practice to switch to ZGR as soon as the temperature gets below 0.5 (same bias, lower variance). Similarly for GR-MC-10 and GR-MC-100 (more computationally costly) the switching points would be t=0.2 and t=0.1, respectively. Thus temperature-scheduled GS-ST would merely play a role of an initialization / warm start to accelerate ZGR.
> > >
> > > At the rebuttal we had a lot of work extending the comparison in other directions, prioritizing direct evaluations (like with like).  Experiments regarding scheduling such as described above are more a matter of applied tuning, however if you find such comparison important (in this or another setup -- please suggest) we would be able to include it in the final version.
> > >
> > > In any case, we do not think that presence or absence of comparison against temperature-scheduled GS should be decisive for our contribution.

---

### Official Review · Reviewer_Waws · 2022-10-25

**Confidence:** 3
**Correctness:** 4
**Technical Novelty And Significance:** 2
**Empirical Novelty And Significance:** 2
**Recommendation:** 6

**Clarity, Quality, Novelty And Reproducibility:**

Clarity
+ The paper is very clear, contextualizing the previous work in the area while accurately differentiating their own work.

Quality
+ The work is relatively high quality. There is a substantial derivation to get the low-$\tau$ behavior of the GR estimator, as well as a thorough experimental evaluation of all of the competing gradient estimators. It is pretty polished.

Novelty
+ The novelty is somewhat limited for me. At this point we have so many estimators for discrete random variables it's unclear what utility is gained by introducing another one. Without a clear argument as to why this estimator's place on the bias-variance curve is uniquely novel, this does not seem very novel overall.

Reproducibiltiy
+ I don't see any provided code, but since the estimator is so simple I can't imagine there would be concerns about reproducibility

**Strength And Weaknesses:**

Strengths
+ The theoretical results are nice. It is very interesting to see that the zero-temperature limit of the GR estimator is given by such a simple combination of the DARN and ST estimator.
+ The experimental results are fairly comprehensive, and presented quite fairly, as far as it is possible to judge.
+ The bias and variance properties of the new estimator are clearly described, and illustrated in the experiments.

Weaknesses
+ The experimental results ultimately do not clearly show superiority of the proposed ZGR estimator. For instance, across table 1 and Figure 2, the Gumbel-Rao estimator performs similarly or better than the ZGR. Similarly in figure 2, the GS-ST(t=2) strictly dominates the ZGR. The experiments do show that the ZGR is a decent enough estimator in practice, but there is still enough variation between all the different estimators that the best bet for the practitioner is to just do a hyperparameter sweep including all the available estimators.
+ Similarly to the previous point, I think there needs to be more discussion about the connection between the bias and variance of the estimator itself, and the error in the optimization process, particularly when combined in with the particular problem of training a neural network with an adaptive optimizer. As the authors point out, simply considering the MSE in the gradient is not necessarily that relevant for the actual question of how well an NN will convergence. For instance, an interesting experiment could be to train the VAE with all the different estimators, using both an adaptive optimizer and regular SGD, to see if the bias might be worse than variance as the authors hypothesise.
+ Minor points:
  + In page 3, shouldn't the CDF of the Gumbel be $e^{-e^{-u}}$?
  + 'unique amongst single-sample categorical estimators' isn't REINFORCE unbiased? Or you need to define what you mean by single-sample estimator
  + Can you not just get in touch with people to get the MNIST dataset you need?

**Summary Of The Paper:**

The paper proposes a gradient estimator for distributions over discrete variables, following in the tradition of the Gumbel-softmax estimator. In this paper, the proposed estimator is the zero-temperature limiting value of the Gumbel-Rao estimator. Interestingly, this is shown to be the same as an average of the DARN and classic Straight-Through (ST) estimator. The authors show that their proposed estimator has zero bias on quadratic objectives, and compare its performance on several downstream tasks.

**Summary Of The Review:**

This paper is a nice piece of work, containing a derivation of the zero-temperature limit of the Gumbel-Rao estimator. However, the experimental comparisons don't currently justify the novelty of the work for me. As it stands, the experiments demonstrate that some gradient estimators are better than others at different tasks, and different combinations of bias and variance are useful for different tasks. This paper provides a new option on the bias-variance curve, but it's not obvious to me that this actually translates to real gains on any downstream task.

Update after the rebuttal:
I'm impressed by the authors' additional experiments and will raise my recommendation to 6.

---

> ### Author Response · Authors · 2022-11-16
> **Discussion**
>
> Thanks for you peer review.
>
> > Experimental results  do not clearly show superiority of the proposed ZGR estimator
>
> Will be addressed by the revision, extending experiments and clarifying practical gains and the knowledge obtained.
>
> > I think there needs to be more discussion about the connection between the bias and variance of the estimator itself, and the error in the optimization process, particularly when combined in with the particular problem of training a neural network with an adaptive optimizer. As the authors point out, simply considering the MSE in the gradient is not necessarily that relevant for the actual question of how well an NN will convergence. For instance, an interesting experiment could be to train the VAE with all the different estimators, using both an adaptive optimizer and regular SGD, to see if the bias might be worse than variance as the authors hypothesise.
>
> At the outset, this question goes out of the scope of the present paper: it is not a subject that we study or make a contribution in. It would be nevertheless of high interest to us understand it better for the future work.
>
> Clearly, of two estimators with the same variance, the one that has a lower bias is better, regardless the optimizer. The same holds for the variance of two estimators with equal bias. Our best grip so far is to measure bias and variance of all estimators experimentally, and we improved in this regard by employing estimates of squared bias and variance that are themselves unbiased.
> If we are to make a tradeoff between bias and variance, the optimal choice apparently depends on the optimizer, its parameters, data and the application in general. At the moment, we do not have an idea how this global effect can be approached theoretically.
>
> We do not see how the proposed experiment could shed some light on the question. By regular SGD do you mean SGD with or without momentum? The gradient is already stochastic due to mini batches and SGD without momentum generally performs worse. Momentum efficiently computes the exponential weighted average of past gradients and thus performs a variance reduction at the expense of some histerezis w.r.t. current training point. We would argue that the hysteresis effect can be interpreted as a bias w.r.t. to the true gradient at the current point. The tradeoff is controlled by the momentum parameter (in Adam it is beta1).  An estimator for categorical variables introduces some additional variance and some additional bias. Suppose we compare performance of all estimators when training with Adam and SGD. What should we compare and how the comparison would imply that e.g. the bias is worse than variance e.g. for the use with Adam? We will appreciate a further discussion on the issue.

---

> > ### Comment · Reviewer_Waws · 2022-11-17
> > **discussion**
> >
> > >> Experimental results do not clearly show superiority of the proposed ZGR estimator
> >
> > >Will be addressed by the revision, extending experiments and clarifying practical gains and the knowledge obtained.
> >
> > I look forward to seeing the new results! Thank you for engaging with this point, which several of the reviewers brought up.
> >
> > > Clearly, of two estimators with the same variance, the one that has a lower bias is better, regardless the optimizer. The same holds for the variance of two estimators with equal bias. Our best grip so far is to measure bias and variance of all estimators experimentally, and we improved in this regard by employing estimates of squared bias and variance that are themselves unbiased. If we are to make a tradeoff between bias and variance, the optimal choice apparently depends on the optimizer, its parameters, data and the application in general. At the moment, we do not have an idea how this global effect can be approached theoretically.
> >
> > I agree in spirit with the claim about preferring, all else being equal, lower bias and variance estimators. But I wouldn't say it's clear that for an arbitrary non-convex optimization like modern neural nets this is the case. For example (off the top of my head), it seems possible to construct a counter-example where the initialization is near a local minimum with a large basin of attraction and the global minimum is further away. A gradient estimator with noise may eventually hit the global minimum but a noise-free one will just converge to the local minimum. Secondly, there are approaches where noise is deliberately added to the gradient: https://arxiv.org/abs/1511.06807. Throw in complications regarding generalization etc and I'd be wary of saying it's clear we want lower variance estimators.
> >
> > However the general point I was trying to make in the review is that since none of the estimators available strictly dominate all the others in terms of bias and variance, we are forced to make a tradeoff between them. Since we have little knowledge of how the (bias,variance) of a gradient estimator translates to quantities we actually care about like (final train loss, final test loss), it's not super useful to know the bias/variance properties of the estimator.
> >
> > Regarding the proposed experiment, I generally just wanted more discussion of the relationship between bias/variance and actual quantities of interest. The experiment I proposed would be one way to discuss this further, but also references to previous work would suffice as well.

---

> > > ### Author Response · Authors · 2022-11-18
> > > **bias-variance**
> > >
> > > > Throw in complications regarding generalization etc and I'd be wary of saying it's clear we want lower variance estimators.
> > >
> > > > Since we have little knowledge of how the (bias,variance) of a gradient estimator translates to quantities we actually care about like (final train loss, final test loss), it's not super useful to know the bias/variance properties of the estimator.
> > >
> > > Interesting thought. I agree that comparing at different trades offs runts into these problems, but I tend to disagree from the methodological point of view. You could as well say that we do not necessarily want to optimize the training loss and you might be right, but this is not constructive.
> > >
> > > The purpose of the gradient estimator is to estimate the gradient, more accurate the better. This steers the rational design of new estimators and is good for optimization whose purpose is to optimize. So far in the literature as well as in our experiments a clear variance or bias reduction always translated to an improvement in the training. And improvement in training translates to at least a better speed and, perhaps with some regularization, to a better test performance. Therefore measuring bias and variance is very important for understanding properties of estimators and designing improvements.
> > >
> > > If there is a extra noise injected for some other purpose than optimization (i.e. in Langevin dynamics to collect samples from posterior), then no idea. But it has to be the right kind of noise, not the DARN estimator noise :)

---

### Official Review · Reviewer_PG9N · 2022-10-29

**Confidence:** 2
**Correctness:** 3
**Technical Novelty And Significance:** 2
**Empirical Novelty And Significance:** 1
**Recommendation:** 3

**Clarity, Quality, Novelty And Reproducibility:**

The paper is written in a rather obscure and indirect language. The state of the hard is comprehensively provided but their key weaknesses and the solution strategy proposed in the paper are not clarified. As a minor point, the acronym of the main contribution of the paper ZGR is defined nowhere in the paper.

**Strength And Weaknesses:**

Strengths:
  i) The studied problem is important for the probabilistic machine learning community
  ii) The paper reports comprehensive experiments that span many critical cases where such an estimator could be useful.

Weaknesses:
 i) The technical novelty is limited.
 ii) Experiment results are weak. ZGR does not appear to show an improvement over for instance ST. I understand that it outperforms the GR family, but what that family can do ST cannot is not clear from the experiments.

**Summary Of The Paper:**

The paper introduces a new estimator called ZGR aimed for the stable and accurate use of discrete random variables as part of approximate inference pipelines.

**Summary Of The Review:**

This is a promising paper, but gives the impression of unfinished work. The proposed method is a very close variant of what already exists. The results are also not strong, although the experiments are comprehensive.

---

> ### Author Response · Authors · 2022-11-07
> **Trying to defend**
>
> While we will be working on the points of criticism that are clear to us (in particular many by Reviewer wSJz) for a revision until Nov 18, we would like to reach some better understanding about other matters. We appreciate your expertise and time in discussing and clarifying this further with us.
>
> > Weaknesses: i) The technical novelty is limited. The proposed method is a very close variant of what already exists. The results are also not strong.
>
> We do not see why such a weakness and what it means. The technical contribution of this paper is focused on a specific problem: analyzing an existing method with the goal of simplification, improvement and understanding the properties. Is this a weakness? The presented theoretical results are presumably novel. Would you expect more / stronger technical results in a single paper?
>
> >Weaknesses: ii) Experiment results are weak. ZGR does not appear to show an improvement over for instance ST. I understand that it outperforms the GR family, but what that family can do ST cannot is not clear from the experiments.
>
> For a range of bias-variance trade-offs ZGR dominates many other estimators and has the simplicity of ST. The GS / GR family appears to be rather popular in the literature with various successful applications which are beyond our capacity to reproduce. We show ZGR is an improvement over ST in VAE while in quantization ST was often better. As you have agreed, ZGR dominates in the GR family in all of the bias variance and computation complexity for a range of temperatures. Overall, our experimental results are in fact an argument in favor of using simpler estimators: ST (if you like) and ZGR = (ST+DARN)/2. A broader evaluation of estimators, including on larger problems, appears to be lacking in the literature and may be a good topic for a separate paper.
>
> At the same time you say “comprehensive experiments that span many critical cases”. If this is the case, our goals in this paper have been achieved. We only tried to get as clear a picture as possible regarding practical properties and performance in at least two diverse application scenarios. The properties demonstrated are to motivate further research on general or specialized estimators.
>
> > The paper is written in a rather obscure and indirect language. The state of the hard is comprehensively provided but their key weaknesses and the solution strategy proposed in the paper are not clarified.
>
> We disagree / do not see this. The weaknesses and the solution strategy are laid out in the introduction.
>
>
> > Correctness: 3: Some of the paper’s claims have minor issues. A few statements are not well-supported, or require small changes to be made correct.
>
> If this grade is not by an accident, we will be happy to fix any minor issues / better support statements. Please let us know of them.
>
> > Technical Novelty And Significance: 2: The contributions are only marginally significant or novel.
>
> This appears to be mostly based on the perceptions of the experiments. What about the following priming, ICLR has in the guides:
>
> "Does it contribute new knowledge and sufficient value to the community? Note, this does not necessarily require state-of-the-art results. Submissions bring value to the ICLR community when they convincingly demonstrate new, relevant, impactful knowledge (incl., empirical, theoretical, for practitioners, etc)."
>
>
> > Empirical Novelty And Significance: 1: The contributions are neither significant nor novel.
>
> Does the paper have no significant value for possible applications and, perhaps more importantly, for the research on the problem of gradient estimation? Should we throw these results away?

---

> > ### Comment · Reviewer_PG9N · 2022-11-28
> > **Keep my grade**
> >
> > Having read the author's response, I decided to keep my original grade. The response does not address any single bit of my concerns.
> >
> > " The technical contribution of this paper is focused on a specific problem: analyzing an existing method with the goal of simplification, improvement and understanding the properties"
> >
> > -- This does not mean the problem or solution approach are novel enough.
> >
> > "This appears to be mostly based on the perceptions of the experiments".
> >
> > --It is not perception but a careful observation of the outcome. The proposed method is simply not advancing the state of the art. The weakness of the experiment results have been raised also by other reviewers.
> >
> > "Does the paper have no significant value for possible applications and, perhaps more importantly, for the research on the problem of gradient estimation?"
> >
> > --This is precisely it. It does not have any significant value for possible applications in the presence of other existing methods that work comparably well.

---

> > > ### Author Response · Authors · 2022-11-28
> > > **The revision and the common response do address all objective concerns**
> > >
> > > > The response does not address any single bit of my concerns.
> > >
> > > Please note that our comment to the initial review above was written in the very beginning of the rebuttal period in the hope that specific concerns and the reference points for the assessment could be clarified to us during the rebuttal. This unfortunately has not happened.
> > >
> > > Our full response, which includes the paper revision, summarized by the global comment https://openreview.net/forum?id=EN8YE5dkOO&noteId=CQFpeVgY99 indeed does address the objective concerns found. Let us check your concerns now:
> > >
> > > >  i) The technical novelty is limited.
> > >
> > > Not true. A theorem is not novel only if it is already known or follows easily from the existing results. Nothing like that has been pointed to us. The review does not seem to evaluate the novelty of the technical work deriving and analyzing ZGR estimator. There is more than one non-trivial and useful theorem.
> > >
> > > > ii) Experiment results are weak. ZGR does not appear to show an improvement over for instance ST.
> > >
> > > Not true. ZGR improves over ST in VAE by a huge margin. Experiment results were substantially extended and show ZGR to be useful.
> > >
> > > > The state of the hard is comprehensively provided but their key weaknesses and the solution strategy proposed in the paper are not clarified.
> > >
> > > The introduction was revised and clarified, with putting more emphasis on the limitations of existing methods, in particular discussing their time complexity.
> > >
> > > > The proposed method is a very close variant of what already exists
> > >
> > > Let us discuss in what sense it is a close variant. We derive it as a limit of GR, however: 1) it is not obvious that such limit would exist, 2) the resulting estimator has nothing in common with the original Gumbel-Softmax relaxation design. The fact that ZGR decomposes as average of ST and DARN (with a special baseline in the later) is not a heuristic but a surprising theoretical implication.
> > >
> > > >  ZGR is defined nowhere in the paper.
> > > Not ture. It was and is defined by eq. (12) or (13). The name it is also defined in the intro by the text: "... we denote this limit estimator as ZGR".
> > >
> > > We see no other (bits of) concerns in the original review.
> > >
> > > > The proposed method is simply not advancing the state of the art. The weakness of the experiment results have been raised also by other reviewers.
> > >
> > > In our view (and this also seems to be ICLR policy as visible from reviewers guide), advancing the state of the art has to be evaluated as 1) bringing new understanding knowledge and tools and 2) improvements in benchmarks / applications / experimental evidence.
> > > We claim important advances of both kinds. Please see the updated experimental results and the paper conclusion. Please discuss with other reviewers if you disagree about the claims in the paper or the evidence in the paper. Please let us know concretely, why the proposed theoretical or experimental results are weak in your opinion.
> > >
> > > > It does not have any significant value for possible applications in the presence of other existing methods that work comparably well.
> > >
> > > Again, this comment focuses on the value for applications, which should not be the single merit for research papers. Furthermore it is not true, because ZGR does make a substantial improvement over the GR family in terms of computation complexity and reducing hyperparameter. The improvement over unbiased estimators is significant because e.g. it is much larger than the improvements demonstrated by e.g. Dong 2021 or Dimitriev and Zhou (2021), which were apparently considered significant enough. The experimental evidence in training quantized models is significant because such evaluation has not been conducted before and it illuminates the role of bias and variance.
> > >
> > > From a professional reviewer we expect to see an argument-based justification of weaknesses and of the acceptance recommendation, not just a subjective view. The present review is rather surficial and, subtracting the weaknesses (which are not true), seems to reduce to just: "the are many estimators available, this is just another one derived from existing ones, not improving in all applications --- why should I care". Sorry for some exaggeration, but clearly on this level it is not possible to distinguish a significant / useful work from insignificant.

---

> > > > ### Comment · Reviewer_PG9N · 2022-11-28
> > > > **Reply**
> > > >
> > > > - "the *acronym* of the main contribution of the paper ZGR was not defined in the Sep 22 version and it is still not defined. I guess it stands for "zero-temperature" GR.
> > > >
> > > > - "the are many estimators available, this is just another one derived from existing ones, not improving in all applications --- why should I care"
> > > >
> > > > This is not what I said. Replace the last part by "not improving in applications reported in the paper with a significant enough margin to conclude that the drawn hypothesis is correct", then you get what I say and this is how the scientific method works.
> > > >
> > > > Let us look at the results tables again:
> > > >    - Table 2 reports training ELBO scores in favor of ZGR, from which I can only conclude that the model fits well to training data. ELBO is a surrogate measure of model fit and what counts is how well the model fits to test data.
> > > >    - Figures 1-2 report some promising results but they are meaningful only if the learned model keeps its performance on the downstream task.
> > > >    - Table 3 reports the first downstream task: MNIST classification. ST outperforms ZGR in almost all cases.
> > > >    - Figure 3 shows that ST also gives smaller estimator variance than ZGR.
> > > >
> > > > I will write another reply only if the authors point me to a factual error in my interpretation of the results tables/figure above, e.g. the score in Fig 3 is indeed not the lower the better but the higher the better, or ZGR's configuration X outperforms all baselines on test data, compare Table X row Y vs Z.

---

> > > > > ### Author Response · Authors · 2022-11-28
> > > > > **Test performance in VAE**
> > > > >
> > > > > Thanks for the question. The evaluation methodology we use in Table 2 is the training performance at a fixed but large enough number of epochs. This is a key performance indicator for comparing gradient estimator in a realistic learning setup. Since we do not perform hyperparameter tuning, the comparison is fair and sound.
> > > > > Furthermore, it seems to be standard in the field to put the training performance forward: e.g.
> > > > > Dong et al. ( NeurIPS 2020) Table 1, Dong et al. (NeurIPS, 2021) Table 1, Dimitriev and Zhou (Neurisp 2021) Table 1
> > > > >  compare by  Train ELBO. And Dimitriev and Zhou (ICML 2021) Table 1, 2 compare by the training log-likelihood.
> > > > >
> > > > > These works additionally verified and illustrated in side results and their appendices that a better training performance translates to a better test performance at least in the benchmarks that use dynamic binarization (binary images are sampled with probabilities equal to pixel values, which has a regularization effect). We therefore did not consider it necessary to do the same. The gap between training and test performance is addressed by regularization and augmentation techniques, which is out of our scope.
> > > > >
> > > > > That said, we do have the respective numbers. Please, **test set negative ELBO** (lower is better):
> > > > >
> > > > > | Dataset       | Method          | C2 V192     | C4 V96      | C16 V48     | C64 V32     |
> > > > > |---------------|-----------------|-------------|-------------|-------------|-------------|
> > > > > | MNIST-B       | GS(t=0.1)       | 94.69±0.06  | 86.8±0.4    | 84.2±0.6    | 87.9±0.9    |
> > > > > |               | GS-ST(t=0.1)    | 94±0.2      | 87±0.3      | 85±0.4      | 90.8±0.5    |
> > > > > |               | GR(t=0.1,K=10)  | 92.5±0.3    | 85.2±0.08   | 82.53±0.07  | 83.7±0.2    |
> > > > > |               | GR(t=0.1,K=100) | 92.5±0.7    | 85.25±0.1   | 82.2±0.4    | 83.2±0.6    |
> > > > > |               | ZGR             | 94.03±0.09  | 86.2±0.2    | 82.5±0.3    | 83.38±0.04  |
> > > > > |               | ST              | 105.3±0.4   | 105.8±0.4   | 106.2±0.3   | 107±0.3     |
> > > > > |               | RF(M=2)         | 97.5±0.3    | 88.9±0.3    | 89.9±0.5    | 97.54±0.08  |
> > > > > |               | RF(M=4)         | 99.1±0.1    | 87.62±0.09  | 84.3±0.5    | 89±0.5      |
> > > > > | Omniglot-28-D | GS(t=0.1)       | 120±0.2     | 118.1±0.2   | 118.93±0.08 | 122.2±0.2   |
> > > > > |               | GS-ST(t=0.1)    | 121.3±0.2   | 118.6±0.1   | 120.26±0.03 | 124.5±0.4   |
> > > > > |               | GR(t=0.1,K=10)  | 118.7±0.4   | 116.8±0.3   | 117.7±0.2   | 119.8±0.3   |
> > > > > |               | GR(t=0.1,K=100) | 118.8±0.1   | 116.36±0.09 | 117.1±0.1   | 118.8±0.2   |
> > > > > |               | ZGR             | 119±0.3     | 116.6±0.2   | 117.3±0.2   | 119±0.2     |
> > > > > |               | ST              | 131.11±0.08 | 131.4±0.1   | 132.15±0.02 | 132.7±0.2   |
> > > > > |               | RF(M=2)         | 123.3±0.4   | 121.1±0.1   | 123.8±0.3   | 128.9±0.3   |
> > > > > |               | RF(M=4)         | 120.6±0.1   | 118.4±0.3   | 120.15±0.04 | 122.84±0.06 |
> > > > >
> > > > >
> > > > > The conclusions are consistent with those we made based on Table 2. We find that ZGR is by a large margin better than ST across all settings, significantly improves over RF(2) and RF(4) overall, especially so with more categories; performs similarly (but cheaper) than GR-MC-100. The latter holds with the exception of MNIST-B C2 and C4 columns. We think because MNIST-B is binarized by a simple threshold, i.e. not dynamically and thus could benefit from adding an extra regularizer.

---

> > > > > > ### Comment · Reviewer_PG9N · 2022-11-28
> > > > > > **Helps tiny bit**
> > > > > >
> > > > > > Thanks for the new results. I take it as my interpretation of all the tables are correct. With the updated version of Table 2, we have four results (corresponding to each item I've listed earlier) in three of which the competitor methods outperform the proposed method. Hence I keep my stance that the experimental evidence in favor of the paper's main claim is weak and stick to my original grade.

---

> > > > > > > ### Author Response · Authors · 2022-11-29
> > > > > > > **No Standing Issues, Contribution**
> > > > > > >
> > > > > > > After having addressed all questions, we see no further issues or problems with the paper. We also know that it has a substantial technical contribution and demonstrates a substantial experimental improvement in VAE. In fact, if we decided not to include the quantization training experiment at all, the paper would demonstrate improvement only. We will not do so because the quantization experiment brings new insights and marks the challenges. In particular, for the first time the variance is measured and discussed in this setting (the bias we were not able to measure accurately, except of the single unit setting, Fig B.3).
> > > > > > >
> > > > > > > The remaining disagreement with the reviewer would probably lie in the plane of the evaluation criteria of scientific work, which, being in unequal positions, we cannot reasonably discuss. We can still address the most recent claims of the reviewer about the paper, but this is becoming progressively pointless as well.
> > > > > > >
> > > > > > > > we have four results (corresponding to each item I've listed earlier) in three of which the competitor methods outperform the proposed method:
> > > > > > > > *  Table 2 reports training ELBO scores in favor of ZGR, from which I can only conclude that the model fits well to training data. ELBO is a surrogate measure of model fit and what counts is how well the model fits to test data.
> > > > > > >
> > > > > > > We provided test results, showing the same relation of methods overall. ZGR outperforms competitor methods in ELBO or is comparable in ELBO and outperforms in speed / gets rid of hyperparameters.
> > > > > > >
> > > > > > > > * Figures 1-2 report some promising results but they are meaningful only if the learned model keeps its performance on the downstream task.
> > > > > > >
> > > > > > > The mentioned figures are related to Table 2, about which we have discussed above, so the preconditioning of the claim is fullfilled.
> > > > > > >
> > > > > > > > * Table 3 reports the first downstream task: MNIST classification. ST outperforms ZGR in almost all cases.
> > > > > > >
> > > > > > > Correct, but we do not see that it would render ZGR useless because in the VAE application the situation is exactly the opposite.
> > > > > > >
> > > > > > > > * Figure 3 shows that ST also gives smaller estimator variance than ZGR.
> > > > > > >
> > > > > > > Yes, and this is a new valuable experimental evidence to us and other researchers who are working on the methods. The bias of ST, in cases where it was possible to measure accurately (Fig. 1, Fig B.3) is larger, so it is a different trade-off. It is rather natural that ZGR and ST, having different trade-offs, outperform each-other in different applications.
> > > > > > >
> > > > > > > > the experimental evidence in favor of the paper's main claim is weak
> > > > > > >
> > > > > > > Our main claim is our whole claim in the abstract (more expanded in the last paragraph of the intro) and in the conclusion. The technical part of the claim appears to be not considered by the reviewer at all. Our conclusions regarding experiments, to the best of our judgement, do not overclaim the experimental evidence.

---

> > > > > > > > ### Comment · Reviewer_PG9N · 2022-11-29
> > > > > > > > **All issues are still standing**
> > > > > > > >
> > > > > > > > „ Correct, but we do not see that it would render ZGR useless because in the VAE application the situation is exactly the opposite.“
> > > > > > > >
> > > > > > > > And this means one cannot draw solid conclusions from the allegedly strong results on the VAE setup. I also do not interpret the reported results on the VAE setting decisive. Low-temperature GR appears to work marginally better there. If two experiment results indicate opposite outcomes, the outcome is inconclusive.
> > > > > > > >
> > > > > > > > „ The mentioned figures are related to Table 2, about which we have discussed above, so the preconditioning of the claim is fullfilled.“
> > > > > > > >
> > > > > > > > No it is not fulfilled. See my comment above. Neither of the results in Table 2, Figures 2-3 and the test ELBOs shown in author response report strong results in favor of the central hypothesis: the advantage of ZGR over other estimators.
> > > > > > > >
> > > > > > > > „ Yes, and this is a new valuable experimental evidence to us and other researchers who are working on the methods. “
> > > > > > > >
> > > > > > > > I really do not understand the relationship of the results here to the main message of the paper, i.e. its alleged value.
> > > > > > > >
> > > > > > > > „  In fact, if we decided not to include the quantization training experiment at all, the paper would demonstrate improvement only. “
> > > > > > > >
> > > > > > > > Then its experiments would have been treated as not comprehensive enough.
> > > > > > > >
> > > > > > > > „ The remaining disagreement with the reviewer would probably lie in the plane of the evaluation criteria of scientific work, which, being in unequal positions, we cannot reasonably discuss. We can still address the most recent claims of the reviewer about the paper, but this is becoming progressively pointless as well.“
> > > > > > > >
> > > > > > > > I fully agree that it has been hard to discuss on a reasonable basis because I am expected to ignore the fact that all empirical results reported by the paper are factually weak. If the main contribution is on the theoretical side and the experiment results do not reflect the value of them, then they are not planned in the way to demonstrate the implications of the theoretical strengths of the proposed approach.

---

> > > > > > > > > ### Author Response · Authors · 2022-12-01
> > > > > > > > > **ZGR is practically useful, experiments are designed well for the theory**
> > > > > > > > >
> > > > > > > > > First of all, thanks a lot for being open to a discussion this far. We would like to address your last concerns by defend the point that ZGR is practically useful.
> > > > > > > > >
> > > > > > > > > * ZGR vs GR-MC(t=0.1, M=100) in VAE (Low-temperature GR appears to work marginally better there).
> > > > > > > > >
> > > > > > > > > Thanks for drawing our attention to this. It happens so that ZGR and GR-MC(t=0.1, M=100) have very similar variance and bias throughout the training process (Fig. 1, Fig B.2 for 16 categories). For configurations with fewer categories, the variance of GR-MC(t=0.1, M=100) is slightly lower than that of ZGR, while bias is still very similar (theoretically and practically). The numbers in Table 2 are from 3 runs and do not necessarily mean a systematic difference. If there is such a systematic effect, it would be due to the difference in the variance. Now, because GR-MC needs significantly more computation, it would be still more beneficial in practice to run ZGR with more samples or simply for more iterations in order to reduce the variance a bit more to compensate.
> > > > > > > > >
> > > > > > > > > Thus in VAE, ZGR achieves the best results while being simpler and faster than GR-MC-100, i.e. it has an advantage over other estimators. We believe this is sufficient to prove usefulness.
> > > > > > > > >
> > > > > > > > > * Quantization
> > > > > > > > >
> > > > > > > > > We have several arguments. First, e.g. Dimitriev and Zhou "CARMS: Categorical-Antithetic-REINFORCE Multi-Sample Gradient Estimator", is a recent work that demonstrated improvement a the single application --- VAE, and this was apparently comprehensive enough (Note also the relative improvement over competing methods is on the edge of statistically significant).
> > > > > > > > > Second, ZGR still outperforms a number of methods in this setting as well. In particular if one forges a hypothetical estimator $\alpha$ ST + (1-\alpha) ZGR (and it does not take much science), it would suffice to check only two values of the hyperparameter $\alpha$ to achieve the best results in both applications with a very simple to implement and cheap method. Finally it is a new experimental evidence, showing that lower variance methods are needed in this application while the bias is not as detrimental.
> > > > > > > > >
> > > > > > > > > >  I am expected to ignore the fact that all empirical results reported by the paper are factually weak
> > > > > > > > >
> > > > > > > > > First we disagree about the premise, it is definitely not "all results are factually weak". They are sufficient to show that the method is practically useful and validate and illustrate the theoretical results. Second, we believe the experiments are designed well enough to verify and illustrate the theory. You would definitely agree that there's more to a paper than the benchmark numbers? Then, is it too much to expect, at least in the spirit, what the ICLR PCs expect from all reviewers? We quote https://iclr.cc/Conferences/2023/ReviewerGuide:
> > > > > > > > > > We ask that: You follow the reviewing guidelines below.
> > > > > > > > > ...
> > > > > > > > >
> > > > > > > > > > What is the significance of the work? Does it contribute new knowledge and sufficient value to the community? **Note, this does not necessarily require state-of-the-art results.** Submissions bring value to the ICLR community when they convincingly demonstrate new, relevant, impactful knowledge (incl., empirical, theoretical, for practitioners, etc).

---

### Official Review · Reviewer_wSJz · 2022-11-02

**Confidence:** 3
**Correctness:** 3
**Technical Novelty And Significance:** 4
**Empirical Novelty And Significance:** 2
**Recommendation:** 8

**Clarity, Quality, Novelty And Reproducibility:**

The paper is well written with minor things that are fixable. The work is solid. Even though the work is incremental on top of ST-GR, but the insights and idea are sharp and useful for the community.

**Strength And Weaknesses:**

Strength:
The paper idea is simple but sharp: inspired by the analysis of Gumbel-Rao (GR) estimator (Paulus et al., 2021) that the estimator performs better at lower temperature, they push the estimator to the zero temperature limit, and found superior performance. Another major contribution is that the paper identifies the connection between GR with Straight-Through (ST) and DARN at the zero temperature limits, for both binary and categorical cases. The insights and analysis are incremental but very important, in terms of both theoretical understanding and improving applicability of the estimator.

Weakness:
There are some minor problems / questions:

1. Questions:
  - In Figure 1, the author compares the computational time across different estimators, is this comparison based on the numbers from different implementations, or are all the estimators implemented in the same code base? Instead of shown the comparison of time, it would be great to include a discussion about complexity analysis on each algorithms, which is not subject to the difference in the implementation and helps the readers / reviewers to better understand how the estimator scales with increasing number of categories. E.g. ARSM requires $\mathcal{O}(C^2)$ evaluations, where $C$ is number of categories.
  - The author(s) discussed the multi-sample version, ZGR s = 2, in experiment section. How does it formulated? Taking average of two estimations from each samples?
  - Is there any analysis for scaling the number of categories? Would the relative performances of different estimators hold for larger / smaller number of categories?
  - MNIST is a standard yet simple task. It would be good to extend the analysis on other benchmarks as well, e.g. FashionMNIST and Omniglot.
  - The author(s) claimed that the bias is very small comparing to the variance, based on the bias and variance estimations at "an evaluation point" "after 100 epochs of training with ARSM". It would be nice if this analysis could be done at an early stage of the training.
  - A side question: instead of setting the temperature to zero or a finite number, how performant is the model if one makes the temperature learnable?

2. Missing baselines:
  - Binary case: as the paper considers the binary case in 3.1. In this case, it's worth to compare against the latest unbiased gradient estimators for binary variables, ARM (Yin et al. 2019) and DisARM (Dong et al. 2020). Both are using antithetical sampling to improve REINFORCE, and DisARM is Rao-Blackwellized ARM by analytically integrating out the randomness.

  - Categorical case: ARSM is a categorical version of ARM. However, previous research (Dong et al. 2021) found ARSM underperform simpler REINFORCE Leave-one-out (RLOO, Kool et al. 2019) baselines by significant amount with comparable number of samples. Also, Dong et al. 2021 proposed Rao-Blackwellized ARSM with binary reparameterization, further improves RLOO.

3. Missing References and things to be fixed:
  - line below Eq (1) "It has proven efficient in practice to ..." Please add reference for this. Also the sentence should be "It has been proven ..."

  - Eq (2), in general, especially for VAE, as the paper considered in experiments, "x" the latent variable, therefore it is usually sampled from posterior distribution. In this case, $\mathcal{L}(x)$ will also depends on $\eta$. Correct derivation should include two terms, e.g. Dong el at. 2021 Eq (2) and relevant discussions, but the term with gradients of $\mathcal{L}(x)$ can typically be estimated with a single Monte Carlo sample and neglected from discussion.

  - Broken sentence: line below Theorem 2: "By quadratic functions we understand quadratic functions .. with arbitraty coefficients"

  - paragraph above 4.2, "GR-MC with K = 10 or K=100 MC samples", it's better to use "M" instead of "K", as the paper used "K" for number of categories.

  - Please try to make the main text self-contained. E.g. Eq (10) mentioned $p_Z(\eta)$, and it is the logistic density. Writing an expression for it, would help the reader to easily reproduce the results.


Reference:
  - Yin, M. and Zhou, M. (2019). ARM: Augment-REINFORCE-merge gradient for stochastic binary networks. ICLR 2029
  - Dong, Z., Mnih, A., and Tucker, G. (2020). DisARM: An antithetic gradient estimator for binary latent variables. NeurIPS 2020
  - Kool, W., van Hoof, H., and Welling, M. (2019). Buy 4 reinforce samples, get a baseline for free! Deep RL Meets Structured Prediction ICLR Workshop.
  - Dong, Z., Mnih, A., and Tucker, G. (2021). Coupled gradient estimators for discrete latent variables. NeurIPS 2021




**Summary Of The Paper:**

The paper presents a variance reduction technique that improves the Straight-Through version of the Gumbel-Softmax estimator by taking the zero-temperature limit. The paper further shows that the proposed estimator could be decomposed into a sum of straight-through estimator and DARN. The superior performance of the proposed estimator using 1 or 2 sample(s) is demonstrated through experiments on MNIST, by comparing against an unbiased categorical estimator ARSM, and Gumbel-Rao with large number of Monte-Carlo samples.



**Summary Of The Review:**

The paper did thorough theoretical analysis on the proposed gradient estimator, with good explanation. The author(s) did good evaluations in experiment section. The reviewer thinks that the experiment section could be improved with better baselines and more analysis on different benchmarks and ablations as mentioned in "Weaknesses" section.
I would raise the score, if the author(s) could help address some of the concerns in Weaknesses section.

---

> ### Author Response · Authors · 2022-11-09
> **Authors response**
>
> We thank the reviewer for a constructive feedback and especially for the update on unbiased estimators.
> This changes the game quite a lot. In a preliminary experiment in our VAE setup we now see that RLOO with 4 samples is on par with ZGR. This is unlike ARSM, which we considered the state of the art: it requires more samples and does not reach that performance. We may conclude that biased estimators, including ZGR, are unnecessary for VAE. We will report this and will try to investigate more the case of multiple stochastic layers, represented in this paper by the relaxed quantization training. Please see below more concretely.
>
> > are all the estimators implemented in the same code base?
>
> We use the same codebase for the whole network, only the categorical layer is specialized. The level of optimization of different estimators is different (as briefly mentioned in the captino of Fig 1): GS is shipped with pytorch and is probably implemented on the C++ level, not the CUDA kernel level. Other estimators are implemented by us on the python level. For GR-MC our implementation is naive by drawing $M$ conditional samples on the forward pass (takes up more memory) but parallel in $M$. We will add a clarification that these time measurements are orientational.
>
> > discussion about complexity analysis on each algorithms
>
> Let $C$ be the number of categories and $L$ the number of (dependent) categorical layers. REINFORCE-like methods still need to learn all parameters (e.g. decoder in VAE), so they still need a full backward pass.
> The complexities as we see it are as follows.
> * ZGR: One forward-backward pass and $O(C)$ per cat. variable.
> * GR-MC with $M$ internal samples: One forward-backward pass and $O(M C)$ per cat. variable.
> * RLOO: $S>=2$ forward passes, at least one backward and $O(C)$ per cat. variable.
> * ARSM: dominated by $S = O(C^2 L^2)$ forward passes.
> * DisARM-IW: $S=2$ forward passes, at least one backward and $O(C^2)$ per cat variable.
>
> ZGR needs to sample once and to compute and differentiate the mean embedding function and the probability of the sample. We think this widens the efficient applicability compared to GS/GR family. E.g., in quantization with triangular noise the complexity $O(1)$ per cat. variable.
>
> A single sample estimate may be a key advantage in some context (e.g. in reinforcement learning in a real environment where rolling back to the same world state to draw more action samples is not possible), but it is not clear to us whether gradient-based estimators are applicable here.
>
> > multi-sample version, ZGR s = 2
>
> This is just uses the average of $s$ independent estimates as a naive variance reduction. We will remove this variant from the main paper and in the appendix will include a separate study how the performance in VAE improves with $s$, in order to illustrate that the variance is a limiting factor. Do you have an idea of a coupled sampling scheme?
>
> > A side question: instead of setting the temperature to zero or a finite number, how performant is the model if one makes the temperature learnable?
>
> As far as we understand, simply considering it as a learnable parameter leads to an invalid method. Yin et a. (2019) ARSM have implemented it in this way, which resulted the performance of GS inferior to ARSM in their Fig.2.
>
> > Missing baselines
>
> We plan to extend the VAE experiment at least to the grid (MNIST, Omniglot) x (batch_size 32, 200) x (categories 2,4,16) x (methods: + RLOO(S=2,4) ) x ( 3-5 random seeds for std estimates).
>
> RLOO will provide a good reference for comparison with further improvements like DisARM-X. We would like to see that ZGR achieves performance of RLOO in this setup and not going to compete with DisARM-X variants (implementing them is not completely trivial). We will describe that these options would be superior to RLOO S=2.
>
> Then we think it will be informative to test and compare to RLOO in the deep setting, represented in our work by the quantization model. If it performs well even in this setting, then biased estimators can go to the dustbin. If not, it will indicate that there is a range of applications for ZGR as it will prove that ZGR is the only cheap method to deliver competitive performance in both VAE and quantization.
>
> > Binary case
>
> In the extension of experiments proposed above, the binary case will be covered in VAE and quantization with 1 bit. However note that ZGR in this case matches the previously known DARN($\frac{1}{2}$). It has been tested by Gu et al. (2016) under the name $\frac{1}{2}$ and in Paulus et al. (2021) (matching to their implementation of FouST -- personal communication with authors). We will thus re-evaluate it, but comparing it further to the specialized binary estimators such as DisARM or ARMS (Dimitriev and Zhou, 2021) is beyond our scope.
>
> In the revision we will of course address the remaining issues as well. Please let us know if you have any follow-up comments at this point, we will be glad to discuss further.

---

> > ### Author Response · Authors · 2022-11-09
> > **Unbiased estimators in the multi-layer / hierarchical case**
> >
> > We have a follow-up question regarding the applicability and complexity of DisARM* and RLOO in a network with multiple stochastic layers such as hierarchical VAE or a deep stochastic network.
> >
> > Namely, Dong et al. (2020) present a variant of binary DisARM for for hierarchical VAEs (Alg. 1. in Appendix C), which requires $O(L^2)$ forward passes as at each layer a new side branch needs to be propagated fully forward.
> >
> > The extension of categorical DisARM-* variants (Dong et al. 2021) to the hierarchical case is not given and not evaluated. We may assume that the variants leveraging binary variables should be extendible in a fashion similar to the binary DisARM. However, such complexity renders them rather impractical for our stochastic quantization application.
> >
> > At the same time, REINFORCE with the leave one out baseline (RLOO) does not necessarily have to follow the same scheme as samples are independent. It seems that 2 global independent samples suffice. Is it known to perform well in the hierarchical case?
> >
> > We would appreciate if the reviewer could comment on these aspects.

---

### Author Response · Authors · 2022-11-18
**Paper Revision**

We are very thankful to reviewers for the feedback and discussion. We've updated the paper, and believe that it substantially improves in the experiments and also in the presentation.

The experiments were extended as follows:
* We compare in VAE and quantization with the REINFORCE leave-one-out (Kool et al. 2019), which is a strong unbiased baseline, denoted RF($M$) in our paper.
* In VAE we evaluate models with different number of categories, including binary; datasets: binary MNIST and Omniglot; confidence intervals. The experiment setup closely follows (Dong et al. 2021) to facilitate indirect comparison.
* We include bias-variance analysis at different snapshots during training in (appendix Fig.B.2)
* A dense range of temperatures is tested for GR-MC in Fig. 2.
* We measure bias and variance in quantization as well in Fig 3.
* Quantization experiment is extended with FashionMNIST dataset (appendix Table B.2) and confidence intervals (with the cross-validation over learning rate, it required about 1000 runs of training).

Other noteworthy improvements:
* Introduction cites recent works on unbiased estimators, is better structured and discusses the main limitations. We summarize the  complexity aspect in Table 1.
* The discussion of experimental results was updated accordingly. Based on the new results we can claim that in VAE ZGR outperforms ST and RF(4) and performs not worse than GR-MC with the best temperature. In quantization it performs decently: it outperforms RF(4), is not worse than GR-MC methods for low temperatures, coming close to ST. Thus, over the two corner applications, ZGR is the only cheap method with a good performance and it can decently replace GR-MC in all tested scenarios, cutting down on computation and hyperparameters.
* The new conclusion clearly states both theoretical and practical gains
* The implementation of ZGR is included in appendix Figure B.1 and we promise in the reproducibility statement to publish all relevant code. We give more details on all implementations, datasets and the experimental setup also in appendix.

---

### Author Response · Authors · 2022-11-21
**Theoretical Complexity**

Unfortunately, there was the following error in the complexity of methods in Table1:

The number of forward passes of unbiased methods ARSM, DisARM, CARMS should be proportional to $L$ and not $L^2$. (but one forward pass still needs to propagate $L$ layers in all methods). The idea was to show separately work needed in terms of convolutions and work needed in terms of handling categorical variables. To give the total computation complexity of all methods, let $F$ be the cost of convolution or a linear transform in a single layer, same in all layers for simplicity. Then the total computation cost of all methods is as follows:

* ST, GS, GS-ST, ZGR: $O(L(F + K))$
* GR-MC(M): $O(L(F + KM))$
* REINFORCE: $O(L(F + K))$
* ARSM: $O(K^2 L^2 F)$
* DisARM-*: $O(2 L^2 (F + K))$
* CARMS(M): $O(M^2 L^2( F + K) + L M^2 K^2)$, where $K^2$ can be reduced to $K$ when the approximation (10) in Dimitriev & Zhou (2021b) is used.

Apologize for the confusion. We will check everything once more in the final version should it be accepted.

---

### Decision · Program_Chairs · 2023-01-20

**Decision:**

Reject

**Justification For Why Not Higher Score:**

The reviews were initially borderline, but after the authors' response and some discussion the overall opinions leaned to the negative, with two reviewers quite negative [3,3] because of the weaknesses mentioned above, one reviewer was marginally positive [6] and wrote "The main concern I have is just the added utility of yet another categorical gradient estimator" and did not defend the paper in the discussion, while the most positive reviewer [8] did not respond during the discussion period.

Ultimately, as I wrote above, this paper does not have one clear case where it is shown the suggested method is practically useful overall, and therefore cannot be published in its current form.

**Justification For Why Not Lower Score:**

N/A

**Metareview: Summary, Strengths And Weaknesses:**

This paper proves the zero temperature limit of the Rao-Blackwellized Straight-Through Gumbel-Softmax Gradient Estimator exists, calculates it, proves a few interesting properties (e.g., being unbiased for quadratic loss functions), finds its relations with other estimators, and performs experiments to validate the method, called ZGR. The reviewers agreed that the theoretical part is rigorous, novel and interesting.

However, they had some concerns regarding its practical significance of this new estimator, because of the experimental results. Specifically, the authors showed results in two areas: VAEs and quantization. In VAEs, the main takeaway was supposed to be that ZGR has similar accuracy as GR-MC, but is x2-x3.5 faster. However, as known empirically and as suggested by Figure B.2, initially it is better to use higher temperatures ("temperature annealing"). Therefore, it is not clear if how significant would be the total acceleration in this VAE case, since it seems it would be beneficial to use ZGR only late in training. The quantization experiments also do not convincingly showcase an instance where ZGR is useful (the aim of this section also seems somewhat orthogonal to the rest of the paper).

Though not every paper should break the state-of-the-art, I believe any paper that proposes a new estimator for usage in ML should at least demonstrate convincingly one ML-related case where the estimator is clearly useful, and this was not done here. I hope the authors can correct this resubmit.